# Rewarded soups: towards Pareto-optimal alignment by interpolating weights fine-tuned on diverse rewards

**Alexandre Rame**[1]*, **Guillaume Couairon**[1,2]†, **Mustafa Shukor**[1]†,
**Corentin Dancette**[1]†, **Jean-Baptiste Gaya**[1,2]†, **Laure Soulier**[1], **Matthieu Cord**[1,3]
[1]Sorbonne Université, CNRS, ISIR, Paris, France    [2]Meta AI    [3]Valeo.ai

## Abstract

Foundation models are first pre-trained on vast unsupervised datasets and then fine-tuned on labeled data. Reinforcement learning, notably from human feedback (RLHF), can further align the network with the intended usage. Yet the imperfections in the proxy reward may hinder the training and lead to suboptimal results; the diversity of objectives in real-world tasks and human opinions exacerbate the issue. This paper proposes embracing the heterogeneity of diverse rewards by following a multi-policy strategy. Rather than focusing on a single a priori reward, we aim for Pareto-optimal generalization across the entire space of preferences. To this end, we propose *rewarded soup*, first specializing multiple networks independently (one for each proxy reward) and then interpolating their weights linearly. This succeeds empirically because we show that the weights remain linearly connected when fine-tuned on diverse rewards from a shared pre-trained initialization. We demonstrate the effectiveness of our approach for text-to-text (summarization, Q&A, helpful assistant, review), text-image (image captioning, text-to-image generation, visual grounding), and control (locomotion) tasks. We hope to enhance the alignment of deep models, and how they interact with the world in all its diversity.

## 1  Introduction

Foundation models [1] have emerged as the standard paradigm to learn neural networks' weights. They are typically first pre-trained through self-supervision [2, 3, 4, 5] and then fine-tuned [6, 7] via supervised learning [8]. Yet, collecting labels is expensive, and thus supervision may not cover all possibilities and fail to perfectly align [9, 10, 11] the trained network with the intended applications. Recent works [12, 13, 14] showed that deep reinforcement learning (DRL) helps by learning from various types of rewards. A prominent example is reinforcement learning from human feedback (RLHF) [12, 15, 16, 17], which appears as the current go-to strategy to refine large language models (LLMs) into powerful conversational agents such as ChatGPT [13, 18]. After pre-training on next token prediction [19] using Web data, the LLMs are fine-tuned to follow instructions [20, 21, 22] before reward maximization. This RL strategy enhances alignment by evaluating the entire generated sentence instead of each token independently, handling the diversity of correct answers and allowing for negative feedback [23]. Similar strategies have been useful in computer vision (CV) [14, 24], for instance to integrate human aesthetics into image generation [25, 26, 27].

**Diversity of proxy rewards.** RL is usually seen as more challenging than supervised training [28], notably because the real reward—ideally reflecting the users' preferences—is often not specified at training time. Proxy rewards are therefore developed to guide the learning, either as hand-engineered metrics [29, 30, 31] or more recently in RLHF as models trained to reflect human preferences

---

*Project lead, main contributor, correspondence to alexandre.rame@isir.upmc.fr.
†Equal experimental contribution, order determined at random.
 Further information and resources related to this project can be found on this website.

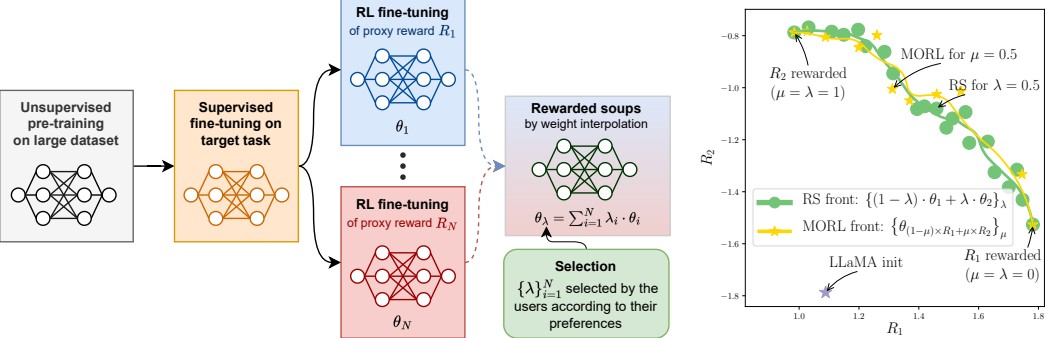

(a) Illustration of our proposed rewarded soup (RS).   (b) LLaMA RLHF for summarization.

Figure 1: Figure 1(a) details the different steps in rewarded soup. After unsupervised pre-training and supervised fine-tuning, we launch $N$ independent RL fine-tunings on the proxy rewards $\{R_i\}_{i=1}^{N}$. Then we combine the trained networks by interpolation in the weight space. The final weights are adapted at test time by selecting the coefficient $\lambda$. Figure 1(b) shows our results (extended in Figure 2(a)) with LLaMA-7b [44] instruct fine-tuned on Alpaca [22], when RL fine-tuning for news summarization [12] with $N = 2$ reward models assessing diverse preferences of summaries. With only two trainings ($R_1$ and $R_2$ rewarded on Figure 1(b)), the $\lambda$-interpolation ($0 \leq \lambda \leq 1$) reveals the green front of Pareto-optimal solutions, i.e., that cannot be improved for one reward without sacrificing the other. RS matches the costly yellow front of multi-objective (MORL) [45, 46] requiring multiple trainings on different linear weightings over the rewards $(1 - \mu) \times R_1 + \mu \times R_2$ with $0 \leq \mu \leq 1$.

[15, 32, 33]. Nonetheless, designing reliable proxy rewards for evaluation is difficult. This *reward misspecification* [9, 34] between the proxy reward and the users' actual rewards can lead to unforeseen consequences [35]. Moreover, the diversity of objectives in real-world applications complicates the challenge. In particular, human opinions can vary significantly [36, 37, 38] on subjects such as aesthetics [39], politics or fairness [40]. Humans have also different expectations from machines: for example, while [41] stressed aligning LLMs towards harmless feedback, [42] requested helpful non-evasive responses, and others' [43] interests are to make LLMs engaging and enjoyable. Even hand-engineered metrics can be in tension: generating shorter descriptions with higher precision can increase the BLEU [29] score but decrease the ROUGE [30] score due to reduced recall.

**Towards multi-policy strategies.** Considering these challenges, a single model cannot be aligned with everyone's preferences [13]. Existing works align towards a consensus-based user [47, 48], relying on the "wisdom of the crowd" [49], inherently prioritizing certain principles [42, 50], resulting in unfair representations of marginalized groups [51, 52]. The trade-offs [53] are decided a priori before training, shifting the responsibility to the engineers, reducing transparency and explainability [54], and actually aligning towards the "researchers designing the study" [13, 55]. These limitations, discussed in Appendix A.1, highlight the inability of single-policy alignment strategies to handle human diversity. Yet, "human-aligned artificial intelligence is a multi-objective problem" [56]. Thus, we draw inspiration from the multi-objective reinforcement learning (MORL) literature [45, 46, 57, 58, 59, 60, 61, 62] and [54]; they argue that tackling diverse rewards requires shifting from single-policy to multi-policy approaches. As optimality depends on the relative preferences across those rewards, the goal is not to learn a single network but rather a **set of Pareto-optimal networks** [63].

In this paper, we propose **rewarded soup** (RS), an efficient and flexible multi-policy strategy to fine-tune any foundation model. As shown in Figure 1(a), we first use RL to learn one network for each proxy reward; then, we combine these expert networks according to user preferences. This a posteriori selection allows for better-informed trade-offs, improved transparency and increased fairness [54, 64]. The method to combine those networks is our main contribution: we do this through **linear interpolation in the weight space**, despite the non-linearities in the network. This is in line with recent findings on linear mode connectivity (LMC) [65, 66]: weights fine-tuned from a shared pre-trained initialization remain linearly connected and thus can be interpolated. This LMC inspired a plethora of weight interpolation (WI) strategies [67, 68, 69, 70, 71, 72], discussed in Section 4. Actually, the name *rewarded soups* follows the terminology of *model soups* [67], as we combine various *ingredients* each rewarded differently. Unlike previous works, which focused on supervised learning, we explore LMC in RL, in a challenging setup where each training run uses a different reward. Perhaps surprisingly, we show that we can trade off the capabilities of multiple weights in a

single final model, thus without any computational overhead. This enables the creation of custom weights for any preference over the diverse rewards. We summarize our contributions as follows:

- We advocate a multi-policy paradigm to align deep generative models with human preferences and reduce reward misspecification.

- We then propose a new multi-policy strategy, rewarded soup, possible when fine-tuning foundation models with diverse rewards. By weight interpolation, it defines a continuous set of (close to) Pareto-optimal solutions, approximating more costly multi-policy strategies.

In Section 3, we consistently validate the linear mode connectivity and thus the effectiveness of RS across a variety of tasks and rewards: RLHF fine-tuning of LLaMA, multimodal tasks such as image captioning or text-to-image generation with diffusion models, as well as locomotion tasks.

## 2 Rewarded soups

### 2.1 RL fine-tuning with diverse rewards

We consider a deep neural network $f$ of a fixed non-linear architecture (e.g., with batch normalization [73], ReLU layers [74] or self-attention [75]). It defines a policy by mapping inputs $x$ to $f(x, \theta)$ when parametrized by $\theta$. For a reward $\hat{R}$ (evaluating the correctness of the prediction according to some preferences) and a test distribution $T$ of deployment, our goal is to maximize $\int_{x \in T} \hat{R}(f(x, \theta))$. For example, with $f$ a LLM, $x$ would be textual prompts, $\hat{R}$ would evaluate if the generated text is harmless [76], and $T$ would be the distribution of users' prompts. Learning the weights $\theta$ is now commonly a three-step process: unsupervised pre-training, supervised fine-tuning, and reward optimization. Yet $\hat{R}$ is usually not specified before test time, meaning we can only optimize a proxy reward $R$ during training. This **reward misspecification** between $R$ and $\hat{R}$ may hinder the alignment of the network with $\hat{R}$. Moreover, the **diversity of human preferences** complicates the design of $R$.

Rather than optimizing one single proxy reward, our paper's first key idea is to consider a family of $N$ diverse proxy rewards $\{R_i\}_{i=1}^N$. Each of these rewards evaluates the prediction according to different (potentially conflicting) criteria. The goal then becomes obtaining a coverage set of policies that trade-off between these rewards. To this end, we first introduce the costly MORL baseline. Its inefficiency motivates our rewarded soups, which leverages our second key idea: weight interpolation.

**MORL baseline.** The standard MORL scalarization strategy [45, 46] (recently used in [62] to align LLMs) linearizes the problem by interpolating the proxy rewards using $M$ different weightings. Specifically, during the *training phase*, $M$ trainings are launched, with the $j$-th optimizing the reward $\sum_{i=1}^N \mu_i^j R_i$, where $\forall j \in \{1, ..., M\}, \{\mu_i^j\}_{i=1}^N \in \Delta_N$ the $N$-simplex s.t. $\sum_{i=1}^N \mu_i^j = 1$ and $0 \leq \mu_i^j \leq 1$. Then, during the *selection phase*, the user's reward $\hat{R}$ becomes known and the $j$-th policy that maximizes $\hat{R}$ on some validation dataset is selected. We typically expect to select $j$ such that $\sum_{i=1}^N \mu_i^j R_i \approx \hat{R}$ linearly approximates the user's reward. Finally, this $j$-th weight is used during the *inference phase* on test samples. Yet, a critical issue is that "minor [preference] variations may result in significant changes in the solution" [77]. Thus, a high level of granularity in the mesh of $\Delta_N$ is necessary. This requires explicitly maintaining a large set of $M \gg N$ networks, practically one for each possible preference. Ultimately, this MORL strategy is unscalable in deep learning due to the **computational, memory, and engineering costs** involved (see further discussion in Appendix A.2).

**Rewarded soup (RS).** In this paper, we draw inspiration from the weight interpolation literature. The idea is to learn expert weights and interpolate them linearly to combine their abilities. Specifically, we propose RS, illustrated in Figure 1(a) and whose recipe is described below. RS alleviates MORL's scaling issue as it requires only $M = N$ trainings while being flexible and transparent.

1. During the *training phase*, we optimize a set of $N$ expert weights $\{\theta_i\}_{i=1}^N$, each corresponding to one of the $N$ proxy rewards $\{R_i\}_{i=1}^N$, and all from a shared pre-trained initialization.

2. For the *selection phase*, we linearly interpolate those weights to define a continuous set of rewarded soups policies: $\{\sum_{i=1}^N \lambda_i \cdot \theta_i\}_{\{\lambda_i\}_{i=1}^N \in \Delta_N}$. Practically, we uniformly sample $M$

interpolating coefficients $\{\{\lambda_i^j\}_{i=1}^N\}_{j=1}^M$ from the $N$-simplex $\Delta_N$ and select the $j$-th that maximizes the user's reward $\hat{R}$ on validation samples, i.e., $\arg\max_{j=1}^M \hat{R}\left(\sum_{i=1}^N \lambda_i^j \theta_i\right)$.

3. For the *inference phase*, we predict using the network $f$ parameterized by $\sum_{i=1}^N \lambda_i^j \theta_i$.

**While MORL interpolates the rewards, RS interpolates the weights.** This is a considerable advantage as the appropriate weighting $\lambda$, which depends on the desired trade-off, can be selected *a posteriori*; the selection is achieved without additional training, only via inference on some samples. In the next Section 2.2 we explicitly state the Hypotheses 1 and 2 underlying in RS. These are considered *Working Hypotheses* as they enabled the development of our RS strategy. Their empirical verification will be the main motivation for our experiments on various tasks in Section 3.

## 2.2 Exploring the properties of the rewarded soups set of solutions

### 2.2.1 Linear mode connectivity of weights fine-tuned on diverse rewards

We consider $\{\theta_i\}_{i=1}^N$ (or $\{\theta_i\}_i$ for brevity) fine-tuned on $\{R_i\}_i$ from a shared pre-trained initialization. Previous works [65, 66, 67, 72] defined linear mode connectivity (LMC) w.r.t. a single performance measure (e.g., accuracy or loss) in supervised learning. We extend this notion in RL with $N$ rewards, and define that the LMC holds if all rewards for the interpolated weights exceed the interpolated rewards. It follows that the LMC condition which underpins RS's viability is the Hypothesis 1 below.

**Working Hypothesis 1** (LMC). $\forall\{\lambda_i\}_i \in \Delta_N$ *and* $k \in \{1, \ldots, N\}$, $R_k(\sum_i \lambda_i \cdot \theta_i) \geq \sum_i \lambda_i R_k(\theta_i)$.

### 2.2.2 Pareto optimality of rewarded soups

The Pareto front (PF) is the set of undominated weights, for which no other weights can improve a reward without sacrificing another, i.e., $\{\theta \mid \nexists \theta' \in \Theta \text{ s.t. } \{R_i(\theta')\}_i >_N \{R_i(\theta)\}_i\}$ where $>_N$ is the dominance relation in $\mathcal{R}^N$. In practice, we only need to retain one policy for each possible value vector, i.e., a Pareto coverage set (PCS). We now introduce the key Hypothesis 2, that state the Pareto-optimality of the solutions uncovered by weight interpolation in RS.

**Working Hypothesis 2** (Pareto optimality). *The set* $\{\sum_i \lambda_i \cdot \theta_i | \{\lambda_i\}_i \in \Delta_N\}$ *is a PCS of* $\{R_i\}_i$.

Empirically, in Section 3, we consistently validate Hypotheses 1 and 2. Theoretically, in Appendix C.2, we prove they approximately hold, in a simplified setup (quadratic rewards with co-diagonalizable Hessians) justifiable when weights remain close.

**Remark 1.** *Hypotheses 1 and 2 rely on a good pre-trained initialization, making RS particularly well-suited to fine-tune foundation models. This is because pre-training prevents the weights from diverging during training [66]. When the weights remain close, we can theoretically justify Hypotheses 1 and 2 (see Appendix C.2) and, more broadly, demonstrate that WI approximates ensembling [78, 79] (see Lemma 4). In contrast, the LMC does not hold when training from scratch [66]. Neuron permutations strategies [80, 81] tried to enforce connectivity by aligning the weights, though (so far) with moderate empirical results: their complementarity with RS is a promising research avenue.*

**Remark 2.** *Pareto-optimality in Hypothesis 2 is defined w.r.t. a set of possible weights $\Theta$. Yet, in full generality, improvements in initialization, RL algorithms, data, or specific hyperparameters could enhance performances. In other words, for real-world applications, the true PF is unknown and needs to be defined w.r.t. a training procedure. In this case, $\Theta$ represents the set of weights attainable by fine-tuning within a shared procedure. As such, in Section 3 we analyze Hypothesis 2 by comparing the fronts obtained by RS and scalarized MORL while keeping everything else constant.*

### 2.2.3 Consequences of Pareto optimality if the user's reward is linear in the proxy rewards

**Lemma 1** (Reduced reward misspecification in the linear case). *If Hypothesis 2 holds, and for linear reward $\hat{R} = \sum_i \hat{\mu}_i R_i$ with $\{\hat{\mu}_i\}_i \in \Delta_N$, then $\exists\{\lambda_i\}_i \in \Delta_N$ such that $\sum_i \lambda_i \cdot \theta_i$ is optimal for $\hat{R}$.*

The proof outlined in Appendix C.1 directly follows the definition of Pareto optimality. In simpler terms, Lemma 1 implies that if Hypothesis 2 holds, RS mitigates reward misspecification for linear rewards: for any preference $\hat{\mu}$, there exists a $\lambda$ such that the $\lambda$-interpolation over weights maximizes the $\hat{\mu}$-interpolation over rewards. In practice, as we see in Figure 5(a), we can set $\lambda = \hat{\mu}$, or cross-validate $\lambda$ on other samples.

# 3 Experiments

In this section we implement RS across a variety of standard learning tasks: text-to-text generation, image captioning, image generation, visual grounding, visual question answering, and locomotion. We use either model or statistical rewards. We follow a systematic procedure. First, we independently optimize diverse rewards on training samples. For all tasks, we employ the default architecture, hyperparameters and RL algorithm; the only variation being the reward used across runs. Second, we evaluate the rewards on the test samples: the results are visually represented in series of plots. Third, we verify Hypothesis 1 by examining whether RS's rewards exceed the interpolated rewards. Lastly, as the true Pareto front is unknown in real-world applications, we present empirical support for Hypothesis 2 by comparing the front defined by RS (sliding $\lambda$ between 0 and 1) to the MORL's solutions optimizing the $\mu$-weighted rewards (sometimes only $\mu = 0.5$ for computational reasons). Implementations are released on github, and this website provides additional qualitative results.

## 3.1 Text-to-text: LLaMA with diverse RLHFs

Given the importance of RLHF to train LLMs, we begin our experiments with text-to-text generation. Our pre-trained network is LLaMA-7b [44], instruction fine-tuned [20, 83] on Alpaca [22]. For RL training with PPO [84], we employ the trl package [85] and the setup from [86] with low-rank adapters (LoRA) [87] for efficiency. We first consider summarization [12, 17] tasks on two datasets: Reuter news [88] in Figures 1(b) and 2(a) and Reddit TL;DR [89] in Figure 2(b). We also consider answering Stack Exchange questions [90] in Figure 2(c), movie review generation in Figure 2(d), and helpfulness as a conversational assistant [49] in Figures 2(e) and 2(f). To evaluate the generation in

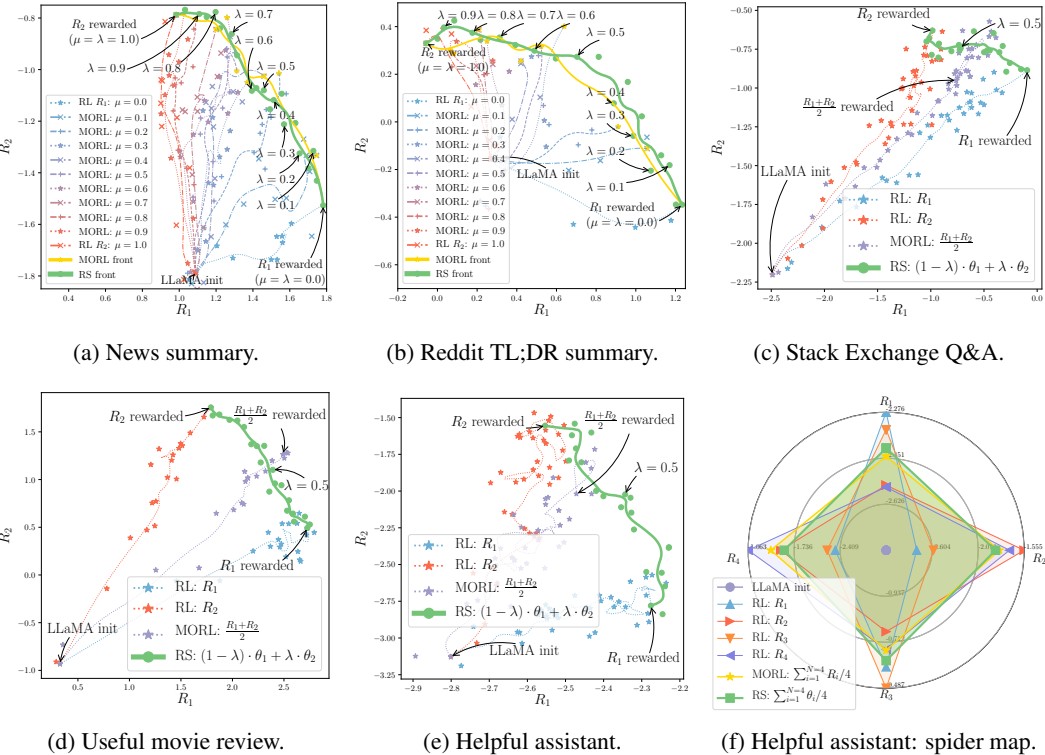

(a) News summary.     (b) Reddit TL;DR summary.     (c) Stack Exchange Q&A.

(d) Useful movie review.     (e) Helpful assistant.     (f) Helpful assistant: spider map.

Figure 2: RLHF results in NLP with LLaMA-7b [44] and reward models $R_i$ from HuggingFace [82]. The blue line reports checkpoints' results along the training trajectory of $\theta_1$ rewarding $R_1$, the red line $\theta_2$ rewarding $R_2$, and the purple line the MORL rewarding $\frac{R_1+R_2}{2}$. Our rewarded soup (RS) linearly interpolates between the weights $\theta_1$ and $\theta_2$; sliding the interpolation coefficient $\lambda$ from 0 to 1 reveals the green solid front of rewarded soups solutions. In Figures 2(a) and 2(b), we additionally show the multiple MORL runs rewarding $(1 - \mu) \times R_1 + \mu \times R_2$ with preferences $0 \leq \mu \leq 1$. It reveals a similar yellow front, yet more costly. In Figure 2(f), we uniformly ($\lambda_i = \frac{1}{4}$) average the weights fine-tuned for the assistant task on $N = 4$ reward models.

the absence of supervision, we utilized $N = 2$ different reward models (RMs) for each task, except in Figure 2(f) where $N = 4$. These RMs were trained on human preferences datasets [15] and all open-sourced on HuggingFace [82]. For example in summarization, $R_1$ follows the "Summarize from Human Feedback" paper [12] and focuses on completeness, while $R_2$ leverages "contrast candidate generation" [91] to evaluate factuality. For other tasks, we rely on diverse RMs from OpenAssistant [92]; though they all assess if the answer is adequate, they differ by their architectures and procedures. Table 1 details the experiments.

The results are reported in Figure 2. The green front, defined by RS between the two weights specialized on $R_1$ and $R_2$, is above the straight line connecting those two points, validating Hypothesis 1. Second, the front passes through the point obtained by MORL fine-tuning on the average of the two rewards, supporting Hypothesis 2. Moreover, when comparing both full fronts, they have qualitatively the same shape; quantitatively in hypervolume [93] (lower is better, the area over the curve w.r.t. an optimal point), RS's hypervolume is 0.367 vs. 0.340 for MORL in Figure 2(a), while it is 1.176 vs. 1.186 in Figure 2(b). Finally, in Figure 2(f), we use $N = 4$ RMs for the assistant task and uniformly average the $N = 4$ weights, confirming that RS can scale and trade-off between more rewards.

## 3.2 Image-to-text: captioning with diverse statistical rewards

RL is also effective for multimodal tasks [14] such as in image captioning [24], to generate textual descriptions of images. Precisely evaluating the quality of a prediction w.r.t. a set of human-written

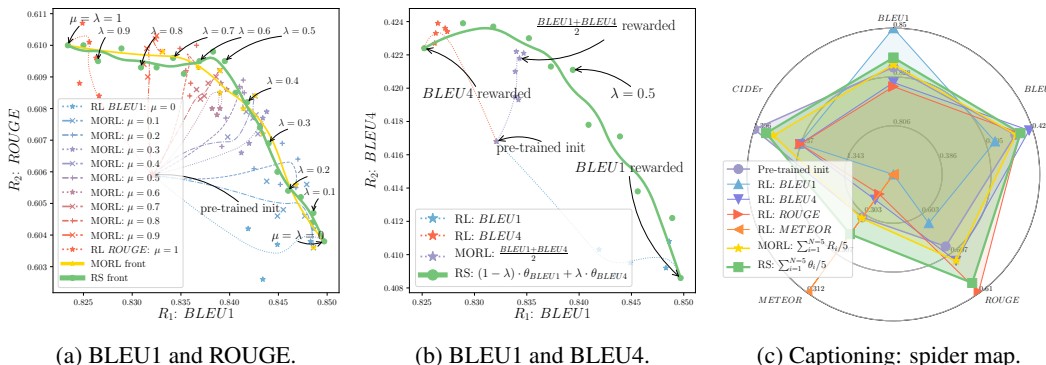

(a) BLEU1 and ROUGE.      (b) BLEU1 and BLEU4.      (c) Captioning: spider map.

Figure 3: Results in image captioning on COCO [94]. As rewards $R_1$ (blue stars every epoch) and $R_2$ (red stars), we consider standard statistical metrics: BLEU1 (1-gram overlap), BLEU4 (4-grams overlap), ROUGE, METEOR and CIDEr. Figure 3(a) include the MORL training trajectories optimizing $(1 - \mu) \times BLEU1 + \mu \times ROUGE$, uncovering a yellow front similar to RS's green front. In Figure 3(c), RS uniformly averages the 5 weights (one for each reward), resulting in the largest area and the best trade-off between the 5 rewards.

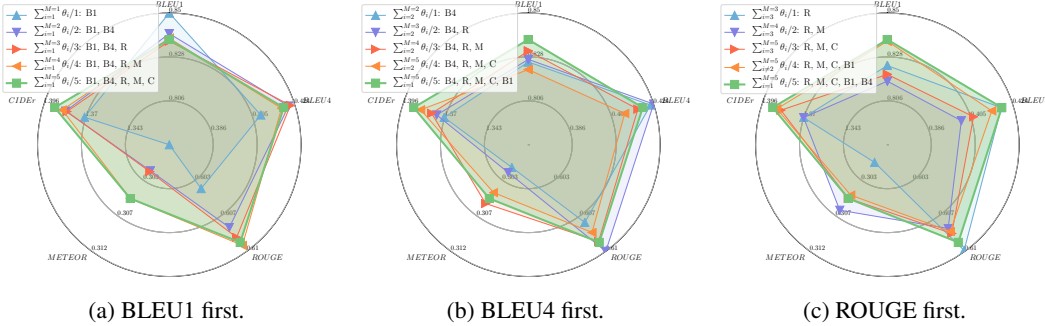

(a) BLEU1 first.      (b) BLEU4 first.      (c) ROUGE first.

Figure 4: Those spider maps uniformly average $1 \le M \le 5$ weights for captioning, where $\theta_1$ is fine-tuned on BLEU1 (B1), $\theta_2$ on BLEU4 (B4), $\theta_3$ on ROUGE (R), $\theta_4$ on METEOR (M) and $\theta_5$ on CIDEr (C). To show different combinations among the $\binom{5}{M}$ possible, we iterate in a clockwise direction starting in Figure 4(a) from $i = 1$ (always including $\theta_1$ optimized on BLEU1), in Figure 4(b) from $i = 2$ (always including $\theta_2$ optimized on BLEU4), and in Figure 4(c) from $i = 3$ (always including $\theta_3$ optimized on ROUGE).

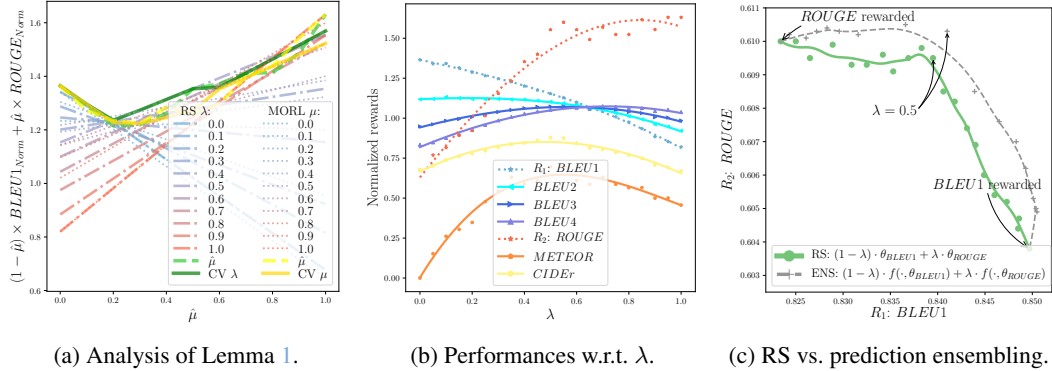

(a) Analysis of Lemma 1.  (b) Performances w.r.t. $\lambda$.  (c) RS vs. prediction ensembling.

Figure 5: Results in captioning for $R_1 = BLEU1$ and $R_2 = ROUGE$. When normalized, rewards are set to 1 for the init and 0 for the worst model. Figure 5(a) validates Lemma 1 by reporting results of RS (for varying $\lambda$) and of MORL (for varying $\mu$) for varying user's preference $\hat{\mu}$. Figure 5(b) evaluates different rewards as a function of the interpolating coefficient. Figure 5(c) reports ensembling scores when interpolating predictions.

captions is challenging, thus the literature relies on various non-differentiable metrics: e.g., the precision-focused BLEU [29], the recall-focused ROUGE [30], METEOR [95] handling synonyms and CIDEr [31] using TF-IDF. As these metrics are proxies for human preferences, good trade-offs are desirable. We conduct our experiments on COCO [94], with an ExpansionNetv2 [96] network and a Swin Transformer [97] visual encoder, initialized from the state-of-the-art weights of [96] optimized on CIDEr. We then utilize the code of [96] and their self-critical [24] procedure (a variant of REINFORCE [98]) to reward the network on BLEU1, BLEU4, ROUGE or METEOR. More details and results can be found in Appendix E.

We observe in Figure 3 that tuning solely BLEU1 sacrifices some points on ROUGE or BLEU4. Yet interpolating between $\theta_1$ and $\theta_2$ uncovers a convex set of solutions approximating the ones obtained through scalarization of the rewards in MORL. When comparing both full fronts in Figure 3(a), they qualitatively have the same shape, and quantitatively the same hypervolume [93] of 0.140. One of the strengths of RS is its ability to scale to any number of rewards. In Figure 3(c), we uniformly ($\lambda_i = \frac{1}{5}$) average $N = 5$ weights fine-tuned independently. It improves upon the initialization [96] and current state-of-the-art on all metrics, except for CIDEr, on which [96] was explicitly optimized. We confirm in Figure 4 that RS can handle more than 2 rewards through additional spider maps. Specifically, we compare the performances across all $N = 5$ metrics when averaging $1 \leq M \leq N$ networks (each fine-tuned on one of the $N$ rewards, thus leaving out $N - M$ rewards at training) and sequentially adding more networks to the weight average. We consistently observe that adding one additional network specialized on one additional reward extends the scope of the possible rewards that RS can tackle Pareto-optimally. Figure 5(a) validates Lemma 1: for any linear preference $\hat{\mu}$ over the proxy rewards, there exists an optimal solution in the set described by RS. Two empirical strategies to set the value of $\lambda$ are close to optimal: selecting $\lambda = \hat{\mu}$ if $\hat{\mu}$ is known, or cross-validating (CV) $\lambda$ if a different data split [99] is available. Moreover, Figure 5(b) (and Appendix E) investigate all metrics as evaluation. Excluding results' variance, we observe monotonicity in both training rewards, linear in BLEU1 and quadratic in ROUGE. For other evaluation rewards that **cannot be linearly expressed** over the training rewards, the curves' concavity shows that RS consistently improves the endpoints, thereby mitigating reward misspecification. The optimal $\lambda$ depends on the similarity between the evaluation and training rewards: e.g., best BLEU2 are with small $\lambda$. Lastly, as per [100] and Lemma 4, Figure 5(c) suggests that RS succeeds because WI approximates *prediction ensembling* [78, 79] when weights remain close, interpolating the predictions rather than the weights. Actually, ensembling performs better, but it cannot be fairly compared as its inference cost is doubled.

### 3.3 Text-to-image: diffusion models with diverse RLHFs

Beyond text generation, we now apply RS to align text-to-image generation with human feedbacks [25, 26, 33]. Our network is a diffusion model [101] with 2.2B parameters, pre-trained on an internal dataset of 300M images; it reaches similar quality as Stable Diffusion [102], which was not used for copyright reasons. To represent the subjectivity of human aesthetics, we employ $N = 2$ open-source

reward models: *ava*, trained on the AVA dataset [103], and *cafe*, trained on a mix of real-life and manga images. We first generate 10000 images; then, for each reward, we remove half of the images with the lowest reward's score and fine-tune 10% of the parameters [104] on the reward-weighted negative log-likelihood [25]. Details and generations for visual inspection are in Appendix F. The results displayed in Figure 6(a) validate Hypothesis 1, as the front described by RS when sliding $\lambda$ from 0 and 1 is convex. Moreover, RS gives a better front than MORL, validating Hypothesis 2. Interestingly, the *ava* reward model seems to be more general-purpose than *cafe*, as RL training on *ava* also enhances the scores of *cafe*. In contrast, the model $\theta_{cafe}$ performs poorly in terms of *ava* in Figure 6(a). Nonetheless, RS with $(1-\lambda) \cdot \theta_{ava} + \lambda \cdot \theta_{cafe}$ outperforms $\theta_{ava}$ alone, not only in terms of *cafe*, but also of *ava* when $\lambda \in \{0.1, 0.2\}$. These findings confirm that RS can better align text-to-image models with a variety of aesthetic preferences. This ability to adapt at test time paves the way for a new form of user interaction with text-to-image models, beyond prompt engineering.

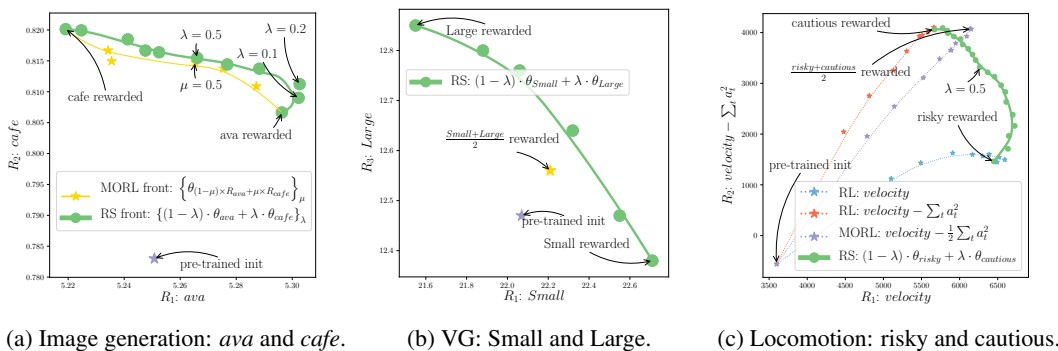

(a) Image generation: *ava* and *cafe*.  (b) VG: Small and Large.  (c) Locomotion: risky and cautious.

Figure 6: Figure 6(a) reports our RLHF experiments on text-to-image generation with diffusion models. From the pre-trained initialization, we learn $\theta_{ava}$ and $\theta_{cafe}$ by optimizing the two reward models *ava* and *cafe*. Interpolation between them reveals the green Pareto-optimal front, above the yellow MORL front. Figure 6(b) report our results in visual grounding (VG) on RefCOCO+ [105], where we optimize to predict boxes with IoU> 0.5 w.r.t. the ground-truth, for objects of either small, medium or large size. Finally, Figure 6(c) report our results from Section 3.5 for the locomotion task with humanoids.

### 3.4 Text-to-box: visual grounding of objects with diverse sizes

We now consider visual grounding (VG) [105]: the task is to predict the bounding box of the region described by an input text. We use UnIVAL [106], a seq-to-seq model that predicts the box as a sequence of location tokens [107]. This model is pre-trained on a large image-text dataset, then fine-tuned with cross-entropy for VG; finally, we use a weighted loss between the cross-entropy and REINFORCE in the RL stage. As the main evaluation metric for VG is the accuracy (i.e., intersection over union (IoU) > 0.5), we consider 3 non-differentiable rewards: the accuracy on small, medium, and large objects. We design this experiment because improving results on all sizes simultaneously is challenging, as shown in Figure 18(c), where MORL performs similarly to the initialization. The results in Figure 6(b) confirm that optimizing for small objects degrades performance on large ones; fortunately, interpolating can trade-off. In conclusion, we can adapt to users' preferences at test time by adjusting $\lambda$, which in turn changes the object sizes that the model effectively handles. On the one hand, if focusing on distant and small objects, a large coefficient should be assigned to $\theta_{Small}$. On the other hand, to perform well across all sizes, we can recover initialization's performances by averaging uniformly (in Figure 18(c)). More details are in Appendix G.

### 3.5 Locomotion with diverse engineered rewards

Teaching humanoids to walk in a human-like manner [108] serves as a benchmark to evaluate RL strategies [109] for continuous control. One of the main challenges is to shape a suitable proxy reward [110, 111], given the intricate coordination and balance involved in human locomotion. It is standard [112] to consider dense rewards of the form $R = velocity - \alpha \times \sum_t a_t^2$, controlling the agent's velocity while regularizing the actions $\{a_t\}_t$ taken over time. Yet, the penalty coefficient $\alpha$ is challenging to set. To address this, we devised two rewards in the Brax physics engine [113]: a risky $R_1$ with $\alpha = 0$, and a more cautious $R_2$ with $\alpha = 1$. Like in all previous tasks, RS's front in

Figure 6(c) exceeds the interpolated rewards, as per Hypothesis 1. Moreover, the front defined by RS indicates an effective balance between risk-taking and cautiousness, providing empirical support for Hypothesis 2, although MORL with $\mu = 0.5$ (i.e., $\alpha = 0.5$) slightly surpasses RS's front. We provide animations of our RL agent's locomotion on our website, and more details are in Appendix H.

### 3.6 Efficiency gain of RS over MORL

The efficiency gain of RS versus MORL is by design; when considering 2 rewards, RS only requires 2 fine-tunings, while MORL actually requires an infinite number of fine-tunings to reveal the entire front of preferences. To end this experimental section, we quantify this efficiency gain by introducing in Figure 7 the expected reward $\mathbb{E}_{\hat{\mu} \sim Unif(0,1)} \hat{R}_{\hat{\mu}}$ where $\hat{R}_{\hat{\mu}} = (1 - \hat{\mu}) \times R_1 + \hat{\mu} \times R_2$ and the expectation is over all the possible user's preferences $\hat{\mu}$. We then measure the difference between the expected rewards for RS (with 2 runs) and MORL (with $M$ runs). Plotting this expected reward advantage for different values of $M$ shows that MORL needs $M \gg 2$ to match RS.

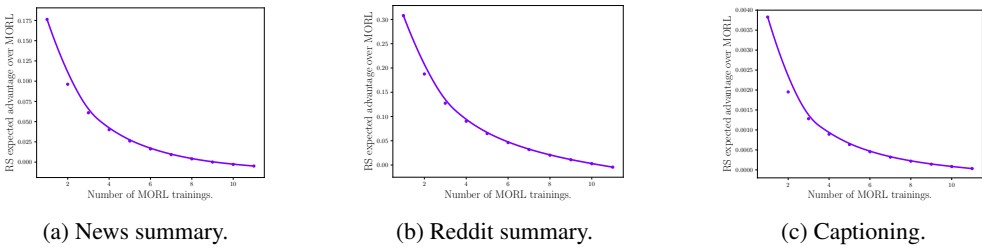

(a) News summary.         (b) Reddit summary.         (c) Captioning.

Figure 7: Expected reward advantage of RS (always requiring only 2 trainings) over MORL (with $M$ trainings), defined as $\mathbb{E}_{\hat{\mu} \sim Unif(0,1)} \left[ max_{\lambda \in \Lambda} \hat{R}_{\hat{\mu}}(\theta_\lambda^{RS}) - \mathbb{E}_{\Lambda_M} \left[ max_{\mu \in \Lambda_M} \hat{R}_{\hat{\mu}}(\theta_\mu^{MORL}) \right] \right]$, where $\hat{R}_{\hat{\mu}} = (1 - \hat{\mu}) \times R_1 + \hat{\mu} \times R_2$ is the user reward for user linear preference $\hat{\mu}$ sampled uniformly between 0 and 1, $\Lambda = \{0, 0.1, ..., 1.0\}$ is the set of the 11 possible values for $\lambda$, and where the expectation for the MORL term is over the $\binom{11}{M}$ possible combinations $\Lambda_M$ of $M$ elements from $\Lambda$ (representing the $M$ linear weightings $\mu$ used for MORL training). We observe that MORL matches RS only for $M$ sufficiently big.

## 4 Related work

Our RS approach leans on two key components from traditional DRL. The first is **proxy rewards**, whose design is challenging. Statistical metrics (the standard in captioning [24]) are not practical to measure human concepts [32] such as helpfulness [49, 76]. Thus recent RLHF works [12, 13, 15] leverage human comparison of prediction to learn a reward model. Second, RS relies on existing **RL algorithms** to maximize the given rewards. RS succeeds with variants of two of the most common, REINFORCE [98] and PPO [84], suggesting it could be applied to others [114, 115]. When dealing with multiple objectives in deep learning, the common strategy is to combine them into a single reward [59, 60]. For example, [116] sum the predictions of a preference RM (as a proxy for helpfulness) and a rule RM (detecting rules breaking); [62] assign different weightings to the relevance/factuality/completeness rewards, thereby customizing how detailed and lengthy the LLMs responses should be. Yet, those **single-policy** approaches (optimizing over a single set of linear preferences) force a priori and uncertain decisions about the required trade-offs [52, 54], as further detailed in Appendix A.1. The **multi-policy** alternatives [45, 46, 57, 58, 61] are not suitable because of the computational costs required to learn set of policies. To reduce the cost, [117, 118, 119, 120] build experts and then train a new model to combine them; [121, 122, 123] share weights across experts; [124, 125, 126, 127] directly train a single model; the recent and more similar work [128] learns one linear embedding per (locomotion) task. Yet, all those works were developed for academic benchmarks [112, 129]; moreover, in terms of Pareto-optimality, they perform equal or worse than the linearized MORL. As far as we know, the only approaches that might improve performances are those inspired from the multitask literature [130], tackling gradients conflicts [131, 132] or different variance scales [133, 134] across tasks. Though they succeed for games such as ATARI [135], our attempts to apply [131] in our setups failed. Overall, as previous MORL works modify the training procedure and usually introduce specific hyperparameters, adapting them to RLHF for foundation

models with PPO is complex; in contrast, RS can be used on top of any RLHF system. Finally, performance and simplicity are not the only advantages of RS over other MORL approaches; in brief, and as discussed in Appendix A.2, RS is compatible with the iterative alignment process.

Recent works extended the **linear mode connectivity** when fine-tuning on different tasks [70, 71, 72, 136], modalities [106] or losses [68, 137], while [138] highlighted some failures in text classification. In contrast, we investigate the LMC in RL. The most similar works are for control system tasks: [139] averaging decision transformers and [140] explicitly enforcing connectivity in subspaces of policies trained from scratch on a single reward. When the LMC holds, combining networks in weights combines their abilities [141, 142]; e.g., averaging an English summarizer and an English-to-French translator can summarize in French [143]. In domain generalization, [67, 68, 144] showed that WI reduces model misspecification [145]; by analogy, we show that RS reduces reward misspecification.

## 5 Discussion: limitations and societal impacts

The recent and rapid scaling of networks presents both opportunities and major concerns [9, 146, 147]. Our approach is a step towards better **empirical alignment** [10, 11]. Yet, many challenges remain untackled. First, proxy rewards may lack robustness [148] or be hacked [149] via adversarial exploitation, making them unreliable. Second, overfitting during training may lead to poor generalization, with a risk of goal misgeneralization [150, 151]. RS could alleviate the impact of some badly shaped proxy rewards and some failed optimizations, as well as tackling Goodhart's law [152]. Yet, without constraint on the test distribution, complete alignment may be impossible [153], for example for LLMs with prompts of arbitrary (long) length.

**Theoretical guarantees** for alignment are also needed [154]. Yet, RS (as all weight interpolation strategies) relies on an empirical finding: the LMC [65], which currently lacks full theoretical guarantees, even in the simplest case of moving averages [100]. That's why we state explicitly our *Working Hypotheses* 1 and 2 in Section 2.2. Nonetheless, we want to point out that in Appendix C.2 we provide theoretical guarantees for the near-optimality of RS when considering quadratic rewards; specifically, in Lemma 3, we bound the reward difference between the optimal policy and our interpolated policy. A remaining limitation is that we theoretically fix issues only for $\hat{R}$ linear over the proxy rewards. Such **linearization** follows the *linear utility functions* setup from the MORL literature [60], that cannot encapsulate all types of (human) preferences [56, 77]. Nonetheless, we showed in Figures 5(b) and 12 that RS improves results even when $\hat{R}$ is not linear. We may further improve results by continually training on new and diverse proxy rewards, to capture the essential aspects of all possible rewards, such that their linear mixtures have increasingly good coverage.

Finally, our a posteriori alignment with users facilitates **personalization** [155] of models. As discussed in Appendix A.1 and in [52], this could increase usefulness by providing tailored generation, notably to under-represented groups. Moreover, the distributed nature of RS makes it parallelizable thus practical in a federated learning setup [156] where data must remain private. Yet, this personalization comes with risks for individuals of "reinforcing their biases […] and narrowing their information diet"[52]. This may worsen the polarization of the public sphere. Under these concerns, we concur with the notion of "personalization within bounds" [52], with these boundaries potentially set by weights fine-tuned on diverse and carefully inspected rewards.

**Conclusion.** As AI systems are increasingly applied to crucial real-world tasks, there is a pressing issue to align them to our specific and diverse needs, while making the process more transparent and limiting the cultural hegemony of a few individuals. In this paper, we proposed rewarded soup, a strategy that efficiently yields Pareto-optimal solutions through weight interpolation after training. Our experiments have consistently validated our working hypotheses for various significant large-scale learning tasks, demonstrating that rewarded soup can mitigate reward misspecification. We hope to inspire further research in exploring how the generalization literature in deep learning can help for alignment, to create AIs handling the diversity of opinions, and benefit society as a whole.

### Acknowledgments

This work was granted access to the HPC resources of IDRIS under the allocations AD011011953R1 and A0100612449 made by GENCI. Sorbonne Université acknowledges the financial support by the ANR agency in the chair VISA-DEEP (ANR-20-CHIA-0022-01).

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

# Rewarded soups: towards Pareto-optimal alignment by interpolating weights fine-tuned on diverse rewards

## Supplementary material

This supplementary material is organized as follows:

The shareable code is released on github. Moreover, you can find additional qualitative results of our experiments on this website.

## A  Discussion

In this section we discuss the benefits of our rewarded soup (RS) approach with respect to the two families of strategies: the **single-policy** and the **multi-policy** approaches.

### A.1  Compared to single-policy approaches

The main reason why single-policy approaches are not suitable is because they optimize over a single set of preferences. In contrast, we build a coverage set of Pareto-optimal policies. This is important for the following reasons, mostly first discussed in Kirk *et al.* [52] and in Hayes *et al.* [54].

Indeed, the user's true reward is highly uncertain before training. This "semi-blind" [54] manual process forces a priori and uncertain decisions about the required trade-offs. It **shifts the responsibility** from the problem stakeholders to the system engineers, who need to anticipate the impact of their choices on the final performance. Critically, the RLHF process may cause the "tyranny of the crowdworker" [52], as models are "tailored to meet the expectations of [...] a small number of crowdworkers primarily based in the US, with little to no representation of broader human cultures, geographies or languages." [52]. Moreover, biased are caused by chaotic engineering choices, and "are exacerbated by a lack of [...] documentation" [52]. In contrast, our approach makes **personalization explicit**, as argued by [52]. Moreover, we could **support decision-making** to find a good balance between (potentially conflicting) parties' interests. This value pluralism [157] can lead to **fairer** and more equitable outcomes [56, 158]. Single-policy cannot adapt to test time requirements; in contrast, RS facilitates personalized assistances [155]. This is all the more important as human preferences change from time to time. In this **dynamic utility function** scenario, RS can quickly adapt with fewer data, by simply adjusting the $\lambda$ to match new preferences (rather than the full network). Finally, RS could also improve the **interpretability** and **explainability** of the decisions. Letting the users decide would make the process more **transparent** [159], which is essential to ensure that the development process is fair, unbiased, and inclusive [160].

### A.2  Compared to multi-policy approaches

The main reason why existing multi-policy approaches through multitasking are not suitable is because of their **computational costs** required to learn a dense set of policies. In contrast, RS only trains the proxy rewards independently and enables the selection of the interpolating coefficient

a posteriori. This is especially useful with large number of rewards and thus growing number of combinations. Second, multitask [130] is challenging; for example, even if the true reward is actually a linear weighted sum of some proxy rewards and those coefficients are known, using those preferences during training can lead to suboptimal results [161], because of conflicting gradients [131, 132] or different variance scales [133, 134]. This has been tackled in RL, but so far mostly for games such as ATARI [135]. Third, our strategy is compatible with the inherent **iterative engineering process** of alignment. Indeed, RS can continually include adjusted opinions while preventing forgetting of the old behaviours. This relates to the **continual learning** challenge, and the empirical observations that weight averaging can reduce catastrophic forgetting [162, 163]. Moreover, as shown in [141] and confirmed in Figure 13(c), negative editing by weight interpolation can fix and force the removal of some behaviours. Finally, RS is computationally effective, requiring **no communication across servers**, thus enabling "embarrassingly simple parallelization" [164]. This facilitates its use in **federated learning** scenario [156] where the data should remain private. Actually, RS follows the **updatable machine learning paradigm** [165], "allowing for the collaborative creation of increasingly sophisticated AI system" [72]. In the future, we may develop open-source personalized models, rewarded on decentralized private datasets, and combine them continuously.

# B  FAQs

We addressed below questions that might arise from readers.

## B.1  What is the difference between rewarded soups and model soups?

Rewarded soups (RS) and model soups (MS) [67] both average weights of models fine-tuned from a shared pre-trained initialization. That's why we chose the same terminology as "model soups" and named our method "rewarded soups". Yet, we want to clarify that RS and MS tackle different problems, have different goals, leading to different methods and implementations.

- RS challenges single-policy approaches to improve alignment in reinforcement learning, and aims at reducing reward misspecification by revealing a Pareto front of solutions across the entire space of preferences: thus RS considers different training objectives for fixed hyperparameters across runs, and non-uniform interpolating coefficients $\lambda$ set a posteriori.

- MS challenges the standard model selection after a grid search to improve generalization in supervised learning, and aims at reducing model underspecification and reducing variance by combining all fine-tuned models: thus MS considers different hyperparameters for a fixed training objective across runs, and (usually) uniform interpolating coefficients $\lambda = \frac{1}{M}$.

These differences mean that MS cannot be applied to reduce reward misspecification, as validated empirically in Figure 13(b) for the captioning task. This Figure 13(b) also shows that RS and MS are actually complementary and can combine their benefits; specifically, reward misspecification and variance reduction.

## B.2  Limitations for the LMC?

### B.2.1  Limitations for the design of networks for the LMC?

In our experiments, we consider different network architectures (transformers, CNNs, and MLPs). We also investigate different training procedures: with low-rank adapters, partial or end-to-end fine-tunings. We do so for many different tasks and modalities: text generation, image captioning, image-to-test generation, visual grounding, etc. Our empirical observation is that, across those setups, the LMC is architecture-agnostic, procedure-agnostic, task-agnostic and modality-agnostic.

The main condition we require is the shared pre-trained initialization [66], so that the weights remain close (as detailed in Remark 1). As a side note, there is another condition suggested by the literature [164, 141]: the LMC would work better when the architecture has enough trainable parameters. For example, according to [141], larger networks may facilitate the orthogonality of the fine-tuned updates; then [141] "speculate that this [orthogonality] enables the combination of task vectors via addition with minimal interference".

### B.2.2 Limitations for the number of training steps for the LMC?

As argued above, good performances are guaranteed when weights remain close; thus longer trainings may be worrisome, as the models may potentially diverge in the weight space. We investigate this question in Figure 8, for the news summarization and the captioning task; we double the number of training steps, and report multiple RS fronts over the course of fine-tuning. Fortunately, we consistently observe good performances for RS along fine-tuning. This confirms that the only condition for the LMC is the shared pre-trained initialization [66].

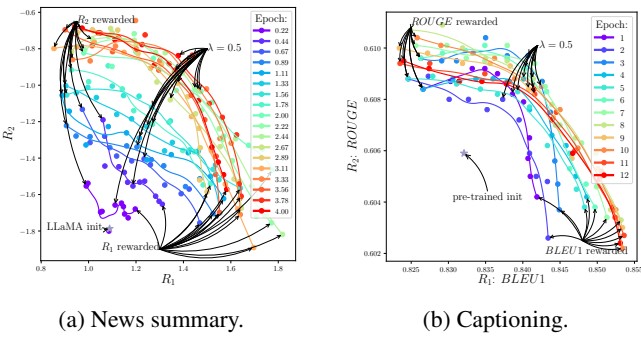

(a) News summary.                    (b) Captioning.

Figure 8: Those figures show how RS's fronts evolve over the course of fine-tuning, and confirms the LMC even when doubling the number of training epochs (previously 2 for news summarization and 6 for image captioning).

### B.2.3 How does the number of rewards (and networks) affects the LMC?

For visualization clarity, the fronts were mostly shown for $N = 2$ rewards, one of the $x$-axis, the other on the $y$-axis. Yet, RS can scale and trade-off between more rewards. We validated this empirically in the spider maps from Figure 2(f) (for text generation), from Figures 3(c) and 4 (for image captioning), and from Figure 18(c) (for visual grounding), where we respectively consider up to $N = 4$, $N = 5$ and $N = 3$ networks fine-tuned on $N$ different rewards, one reward each.

### B.3 Comparison of MORL and RS

### B.3.1 How to evaluate Pareto-optimality?

Given a fixed preference $\hat{\mu}$ between two rewards $R_1$ and $R_2$, we would like to compare our RS policy to an oracle policy maximizing $(1 - \hat{\mu}) \times R_1 + \mu \times R_2$ in test. Yet, this oracle policy (and the true Pareto front) is unknown in real-world applications.

That's why, in practice, and as argued in Remark 2, we presented empirical support for Hypothesis 2 by considering the MORL's solutions fine-tuned to optimize $(1 - \hat{\mu}) \times R_1 + \mu \times R_2$ in train, for $0 \leq \mu \leq 1$. In other words, the linearized MORL is our reference to evaluate Pareto optimality. Overall, in Section 3, MORL and RS usually perform similarly (with small differences further discussed below in Appendices B.3.2 and B.3.3). Our conclusion is that rewarded soup is an empirical solution **towards** Pareto-optimality, with indeed an experimental limitation highlighted in the paper's name.

### B.3.2 How does reward diversity affect the effectiveness of RS?

Our experiments in captioning and image generation provide empirical evidence that the more similar the rewards, the higher the gains of RS versus MORL.

In the captioning experiment, by analyzing the transfer abilities across rewards in the spider maps from Figure 3(c), we can deduce that BLEU4 and ROUGE are more similar than BLEU1 and ROUGE, while METEOR is an outlier (fine-tuning on METEOR worsens the results for the other rewards). Then, we can observe that the gains of RS versus MORL are consistent with these similarities across rewards. Specifically, when considering $R_2 = ROUGE$, the RS green front is more convex and significantly above the MORL yellow front in Figure 11(a) (with $R_1 = BLEU4$) than in Figure 3(a) (with $R_1 = BLEU1$). In Figure 12(b), with $R_2 = METEOR$, MORL performs better than RS.

Similarly, in the image generation experiment, when we consider two (arguably similar) aesthetic rewards in Figure 6(a) to fine-tune a diffusion model, RS's front is to the right and above MORL's front. In contrast, performances get worse in Figure 14 where we also include an *nsfw* reward inversely correlated with image quality.

In conclusion, despite using diverse and heterogeneous rewards that are in tension, we consistently obtain positive results. Yet, in the case where rewards are fully antagonist, we acknowledge that RS is likely to produce less favorable results. This empirical limitation of weight interpolation can be explained in two different ways. (i) Intuitively from a loss landscape perspective: weights fine-tuned on antagonist rewards will be more distant, thus potentially breaking the linear mode connectivity. (ii) Theoretically thanks to Lemma 3, where we bound the difference between the optimal reward and RS's reward by a RHS term growing the maximum of eigenvalues ratio for rewards' Hessians: if the rewards are more diverse, their Hessians would have more different eigenvalues, thus maximum of eigenvalues ratio would grow, the RHS term would grow in Lemma 3, and our guarantees for the optimality of RS would get loose.

As a final note, to tackle this limitation under antagonist rewards, the complementarity of MORL and RS appears as a promising research direction; this is further discussed in the legend of Figure 13(a) for the captioning task and in Appendix F.2 for the image generation task.

### B.3.3  Why RS is sometimes superior to MORL?

We observe a few times that the RS solutions are actually above the linearized MORL solutions. We speculate this is related to the multiple benefits of weight interpolation. The main benefit that we discuss in our paper is the ability to interpolate between different policies: from this benefit, we would expect RS to perform similarly to MORL. The second benefit from weight averaging is the implicit regularization, causing variance reduction and stabilizing performances [100, 144]. This is the main focus of the traditional weight averaging literature, for example in model soups [67]. In conclusion, we speculate that this second benefit (combined with the first) can explain why RS sometimes outperforms MORL.

## C Theoretical insights

### C.1 Proof of Lemma 1

*Proof.* Considering $\theta$ maximizing $\hat{R}$, we first show that $\theta$ is on the PF of $\{R_i\}_i$. Otherwise, considering $\theta' >_N \theta$ and as $\forall i, \hat{\mu}_i \geq 0$, we have $\sum_i \hat{\mu}_i R_i(\theta') > \sum_i \hat{\mu}_i R_i(\theta)$. This implies that $\theta'$ would produce a better policy than $\theta$ for $\hat{R} = \sum_i \hat{\mu}_i R_i$ and thus the contradiction. Finally, as $\theta$ is on the PF and by definition of a PCS, there exists $\lambda$ s.t. $\forall k, R_k(\sum_i \lambda_i \cdot \theta_i) = R_k(\theta)$. $\square$

### C.2 Theoretical guarantees with quadratic rewards

In this section, we provide theoretical guarantees for the near-optimality of RS when considering quadratic rewards. This simplification amounts to replacing the rewards by their second-order Taylor approximation, which is a realistic assumption when the weights remain within a small neighborhood.

#### C.2.1 Simple case with Hessians proportional to the Identity matrix

For the first Lemma 2, we make the following simplifying Assumption 1.

**Assumption 1** (Hessians proportional to the Identity matrix.). *Every reward $R_i$ is quadratic, with Hessians proportional to $\mathbb{I}_d$. Specifically, let $\Theta \subset \mathbb{R}^d$ be the set of possible weights, and let $\{R_i\}_{i=1}^N$ be the $N$ rewards, we can write for $i \in \{1, ..., N\}$:*

$$\forall \theta \in \Theta, \quad R_i(\theta) = R_i(\theta_i) - \eta_i \|\theta - \theta_i\|^2 \tag{1}$$

*where $\eta_i \in \mathbb{R}_+^*$ and $\theta_i$ is the global maximum for reward $R_i$.*

**Lemma 2.** *Let $\hat{\mu} = (\hat{\mu}_1, ..., \hat{\mu}_N) \in \Delta_N$. Then, under Assumption 1, the reward $R_{\hat{\mu}} = \sum_i \hat{\mu}_i \times R_i$ is maximized on the convex hull of $\{\theta_1, ..., \theta_N\}$.*

*Proof.* The function $R_{\hat{\mu}}$ is quadratic thus has an unique global maximum $\hat{\theta}$, that we find analytically:

$$\nabla_\theta R_{\hat{\mu}}(\hat{\theta}) = 0 \implies \sum_{i=1}^N \mu_i \eta_i \cdot (\hat{\theta} - \theta_i) = 0$$

$$\implies \hat{\theta} = \frac{\sum_{i=1}^N \hat{\mu}_i \eta_i \cdot \theta_i}{\sum_{i=1}^N \hat{\mu}_i \eta_i}$$

Since all the $\hat{\mu}_i \eta_i$ are positive or zero, and at least one is greater than zero, $\hat{\theta}$ is indeed in the convex hull of $\{\theta_1, ..., \theta_N\}$. $\square$

**Remark 3.** *Under Assumption 1, the reward functions are concave; thus we can reasonably assume that each fine-tuning procedure for $R_i$ reaches its global optimum $\theta_i$ for $i \in \{1, ..., N\}$. Then, Lemma 2 tells us that the maximum value for linear user's reward $R_{\hat{\mu}}$ is obtainable by weight interpolation between the $\{\theta_i\}_{i=1}^N$: the interpolating coefficients in $\Delta_N$ such that $\lambda_i \propto \hat{\mu}_i \eta_i$ make rewarded soups optimal.*

#### C.2.2 Advanced case with diagonal Hessians

We now consider the more complex case with the relaxed Assumption 2. For simplicity, we only consider $N = 2$ rewards $R_1$ and $R_2$.

**Assumption 2** (Diagonal Hessians). *The rewards are quadratic, with Hessians diagonal negative definite. Specifically, we can write for $i \in \{1, 2\}$:*

$$\forall \theta = (\theta^1, ..., \theta^d) \in \Theta, \quad R_i(\theta) = R_i(\theta_i) - \sum_{j=1}^d \eta_i^j (\theta^j - \theta_i^j)^2, \tag{2}$$

*where $(\eta_i^1, ... \eta_i^d) \in \{\mathbb{R}_+^*\}^d$ and $\theta_i = (\theta_i^1, ..., \theta_i^d)$ is the global maximum for reward $R_i$.*

**Remark 4.** *This diagonal Assumption [2] of the Hessian is common: for example in optimization [166, 167], to prune networks [168] or in out-of-distribution generalization [169]. This strong assumption is supported by the empirical observation [170] that Hessians are diagonally dominant, in particular at the end of training. Also, we note that our findings remain valid assuming only that the Hessians are co-diagonalizable.*

**Lemma 3.** *We consider the user's reward $R_{\hat{\mu}} = (1 - \hat{\mu}) \times R_1 + \hat{\mu} \times R_2$ with $\hat{\mu} \in [0, 1]$, and*

$$\Delta R_{\hat{\mu}} = \max_{\theta \in \Theta} R_{\hat{\mu}}(\theta) - \max_{\lambda \in [0,1]} R_{\hat{\mu}}((1 - \lambda) \cdot \theta_1 + \lambda \cdot \theta_2). \tag{3}$$

*$\Delta R_{\hat{\mu}}$ corresponds to the difference in terms of $R_{\hat{\mu}}$ between the global maximum and the maximum reachable by weight interpolation through rewarded soups (with a single interpolating coefficient for all dimensions). Then, under Assumption [2], we have:*

$$\Delta R_{\hat{\mu}} \leq \frac{\hat{\mu}^2(1 - \hat{\mu})^2(M\Delta_1 - \Delta_2)(M\Delta_2 - \Delta_1)}{(\hat{\mu}(1 - \hat{\mu})(M - 1)^2 + M)((1 - \hat{\mu})\Delta_1 + \hat{\mu}\Delta_2)}, \tag{4}$$

*where $M = \max_{j \in \{1,...,d\}} \max\left(\frac{\eta_1^j}{\eta_2^j}, \frac{\eta_2^j}{\eta_1^j}\right)$ is the maximum of eigenvalues ratio, $\Delta_1 = R_1(\theta_1) - R_1(\theta_2)$ and $\Delta_2 = R_2(\theta_2) - R_2(\theta_1)$.*

*When $\Delta_1 = \Delta_2$, the bound simplifies into:*

$$\Delta R_{\hat{\mu}} \leq \frac{\hat{\mu}^2(1 - \hat{\mu})^2(M - 1)^2}{\hat{\mu}(1 - \hat{\mu})(M - 1)^2 + M} \Delta_1 \tag{5}$$

*Furthermore, when the Hessians are equal, then $M = 1$ and $\Delta R_{\hat{\mu}} = 0$: RS is optimal .*

*Proof.* This novel proof is in three steps. First, we find $\hat{\theta}$ maximizing $R_{\hat{\mu}}(\theta)$ for $\theta$ on the full set of weights $\Theta$. Second, we find $\bar{\lambda}$ maximizing $R_{\hat{\mu}}((1 - \lambda) \cdot \theta_1 + \lambda \cdot \theta_2)$ for $\lambda \in [0, 1]$ and thus defining the best interpolation between the expert weights. Finally, we bound $\Delta R_{\hat{\mu}}$, the differences between their rewards, by applying the Bhatia-Davis inequality.

**First step.** Let's first find the maximum of $R_{\hat{\mu}}$ on $\Theta$. Denoting $S = (1 - \hat{\mu}) \times R_1(\theta_1) + \hat{\mu} \times R_2(\theta_2)$, we have for all $\theta \in \Theta$:

$$R_{\hat{\mu}}(\theta) = S - \sum_{j=1}^{d} \left((1 - \hat{\mu})\eta_1^j\left(\theta^j - \theta_1^j\right)^2 + \hat{\mu}\eta_2^j\left(\theta^j - \theta_2^j\right)^2\right) \tag{6}$$

Since $R_{\hat{\mu}}$ is a sum of concave quadratic functions, it has a unique global maximum reached at a point we note $\hat{\theta} = \left(\hat{\theta}^1, ..., \hat{\theta}^d\right)$. The global maximum can be computed by differentiating $R_{\hat{\mu}}$ with respect to each variable $\theta^j$, which gives:

$$\hat{\theta}^j = \left(1 - \hat{\lambda}^j\right) \cdot \theta_1^j + \hat{\lambda}^j \cdot \theta_2^j$$

where the interpolating coefficients per dimension $\hat{\lambda}^j$ are defined for $j \in \{1, ..., d\}$ as:

$$\hat{\lambda}^j = \frac{\hat{\mu}\eta_2^j}{(1 - \hat{\mu})\eta_1^j + \hat{\mu}\eta_2^j} \in [0, 1]. \tag{7}$$

**Second step.** With $\lambda \in [0, 1]$ and $\theta = (1 - \lambda) \cdot \theta_1 + \lambda \cdot \theta_2$, we can write $R_{\hat{\mu}}(\theta)$ as a function of $\lambda$:

$$R_{\hat{\mu}}(\theta) = S - \sum_{j=1}^{d}\left(\left((1 - \hat{\mu})\eta_1^j + \hat{\mu}\eta_2^j\right)\left(\lambda - \hat{\lambda}^j\right)^2 + \frac{\hat{\mu}(1 - \hat{\mu})\eta_1^j\eta_2^j}{(1 - \hat{\mu})\eta_1^j + \hat{\mu}\eta_2^j}\right)\left(\theta_1^j - \theta_2^j\right)^2$$

$$= R_{\hat{\mu}}(\hat{\theta}) - \sum_{j=1}^{d} p_j\left(\lambda - \hat{\lambda}^j\right)^2 \tag{8}$$

where $p_j$ is defined as $p_j = \left((1 - \hat{\mu})\eta_1^j + \hat{\mu}\eta_2^j\right)\left(\theta_1^j - \theta_2^j\right)^2$.

From Equation (8), we can compute the maximum reward obtainable for weight averaging $\max_{\lambda \in [0,1]} R_{\hat{\mu}}((1 - \lambda) \cdot \theta_1 + \lambda \cdot \theta_2)$. Since the function $\lambda \mapsto R_{\hat{\mu}}((1 - \lambda) \cdot \theta_1 + \lambda \cdot \theta_2)$ is a concave quadratic function, there is a unique value $\bar{\lambda}$ maximizing $R_{\hat{\mu}}$ equal to

$$\bar{\lambda} = \frac{\sum_{j=1}^{d} p_j \hat{\lambda}^j}{\sum_{j=1}^{d} p_j}. \tag{9}$$

Since all $p_j$ are positive and all $\hat{\lambda}^j$ are between 0 and 1, $\bar{\lambda}$ is also between 0 and 1. Therefore, $R_{\hat{\mu}}((1 - \bar{\lambda}) \cdot \theta_1 + \bar{\lambda} \cdot \theta_2)$ is indeed the maximum reward for rewarded soups.

**Third step.** Applying Equation (8) to $\bar{\lambda}$ gives:

$$\Delta R_{\hat{\mu}} = R_{\hat{\mu}}(\hat{\theta}) - R_{\hat{\mu}}((1 - \bar{\lambda}) \cdot \theta_1 + \bar{\lambda} \cdot \theta_2) \tag{10}$$

$$= \sum_{j=1}^{d} p_j \left( \bar{\lambda} - \hat{\lambda}^j \right)^2 \tag{11}$$

$$= \left( \sum_{j=1}^{d} \frac{p_j}{\sum_{i=1}^{n} p_i} \left( \bar{\lambda} - \hat{\lambda}^j \right)^2 \right) \left( \sum_{j=1}^{n} p_j \right) \tag{12}$$

The second term in Equation (12) can be simplified as:

$$\sum_{j=1}^{d} p_j = (1 - \hat{\mu})\Delta_1 + \hat{\mu}\Delta_2. \tag{13}$$

The core component of this proof is the upper bounding of the first term in Equation (12). The key idea is to recognize the variance of a discrete random variable $\Lambda$ with $\mathbb{P}(\Lambda = \hat{\lambda}_i) = \frac{p_i}{\sum_{j=1}^{n} p_j}$; then, $\bar{\lambda}$ from Equation (9) is actually the expectation of $\Lambda$. Then, we can apply the **Bhatia-Davis inequality**, as recalled in Equation (14), on the variance of a bounded random variable $a \leq \Lambda \leq b$:

$$Var(\Lambda) \leq (b - \mathbb{E}(\Lambda))(\mathbb{E}(\Lambda) - a) \tag{14}$$

Therefore Equation (12) is bounded by:

$$\Delta R_{\hat{\mu}} \leq \left( \max_{1 \leq j \leq d} \hat{\lambda}^j - \bar{\lambda} \right) \left( \bar{\lambda} - \min_{1 \leq j \leq d} \hat{\lambda}^j \right)((1 - \hat{\mu})\Delta_1 + \hat{\mu}\Delta_2). \tag{15}$$

Now, we bound the variables $\hat{\lambda}^j$, since $1/M \leq \eta_1^j / \eta_2^j \leq M$. Then for all $j$ we have:

$$\frac{\hat{\mu}}{(1 - \hat{\mu})M + \hat{\mu}} \leq \hat{\lambda}^j \leq \frac{\hat{\mu}M}{(1 - \hat{\mu}) + \hat{\mu}M}, \tag{16}$$

and thus:

$$\Delta R_{\hat{\mu}} \leq \left( \frac{\hat{\mu}M}{1 + \hat{\mu}(M - 1)} - \bar{\lambda} \right) \left( \bar{\lambda} - \frac{\hat{\mu}}{M - \hat{\mu}(M - 1)} \right)((1 - \hat{\mu})\Delta_1 + \hat{\mu}\Delta_2). \tag{17}$$

Finally, noting that $\Delta_i = \sum_{j=1}^{d} \eta_i^j \left( \theta_2^j - \theta_1^j \right)^2$, we deduce from Equation (9) that $\bar{\lambda} = \frac{\hat{\mu}\Delta_2}{(1 - \hat{\mu})\Delta_1 + \hat{\mu}\Delta_2}$. Replacing this in the previous Equation (17) gives the final Equation (4), concluding the proof. $\square$

**Remark 5.** *As a final remark, please note that the suboptimality of RS comes from the need of having one single interpolating coefficient $\bar{\lambda}$ for all $d$ parameters $(\theta^1, \ldots, \theta^d)$ of the network. Yet, the advanced merging operations in [69] remove this constraint, with interpolating coefficients proportional to the eigenvalues of the Fisher matrices [171], which actually approximate the eigenvalues of the Hessian [172, 173]. Combining [69] and our RS is a promising research direction, the key issue being the computation of the Fisher matrices [174] for networks with billions of parameters.*

### C.2.3 Bound visualization

We visualize in Figure 9 the bound given by Lemma 3. We show that for small values of $M$ like $M = 2$, the value of $R_{\hat{\mu}}$ for RS is quite close to the global optimum. Also, recall that RS theoretically matches this upper bound when $M = 1$. For larger values like $M = 10$, the bound is less tight, and we note that the maximum value of $R_{\hat{\mu}}$ approaches the constant function 1 as $M \to \infty$.

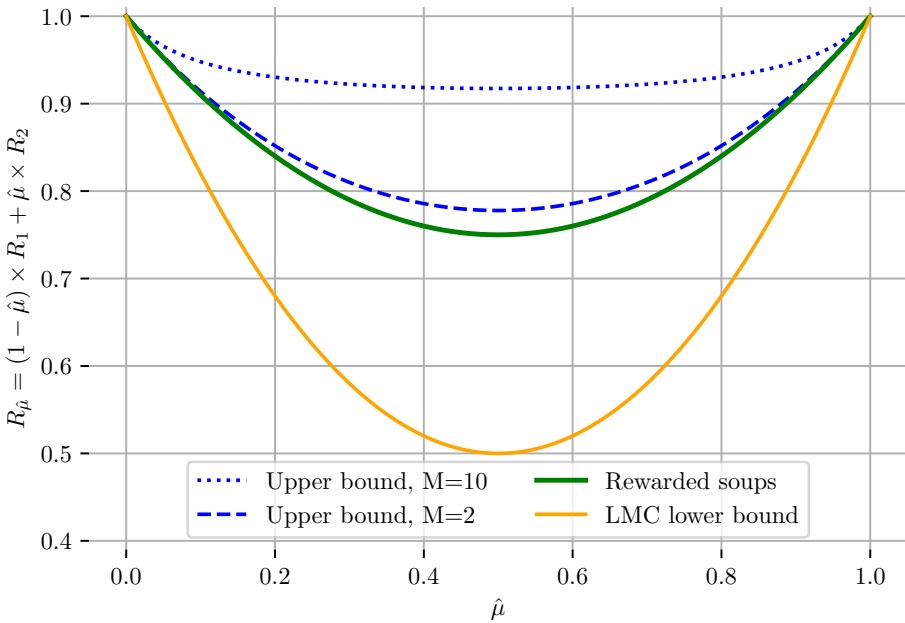

Figure 9: Illustration of the bound given by Lemma 3 under Assumption 2. For simplicity, we showcase the case where $R_1(\theta_1) = R_2(\theta_2) = 1$, $R_1(\theta_2) = R_2(\theta_1) = 0$, thus $\Delta_1 = \Delta_2 = 1$. In green, we plot the rewards obtained with rewarded soups for the optimal $\bar{\lambda}$, i.e., $R_{\hat{\mu}}\left((1 - \bar{\lambda}) \cdot \theta_1 + \bar{\lambda} \cdot \theta_2\right)$, whose value is independent of $M$ in this case. In blues, we plot the maximum value of $\mathcal{R}_{\hat{\mu}}$ given by Equation (5) in Lemma 3, for $M = 2$ and $M = 10$. For reference, we also plot the values for the lower bound in the LMC Hypothesis 1, i.e., equal to $(1 - \hat{\mu})(1 - \bar{\lambda})R_1(\theta_1) + \hat{\mu}\bar{\lambda}R_2(\theta_2)$. As RS outperforms this lower bound, it validates Hypothesis 1 in this case.

### C.3 Similarity between weight interpolation and functional ensembling

**Lemma 4** ($\lambda$-interpolation of weights approximates the $\lambda$-ensembling of predictions. Adapted from [67, 68, 100].)**.** *Given $\theta_1$ and $\theta_2$ optimized for $R_1$ and $R_2$ s.t. they remain close, i.e., $\|\theta_1 - \theta_2\|_2 \approx 0$. Denoting $\theta_\lambda$ the interpolated weights $\theta_\lambda = (1 - \lambda) \cdot \theta_1 + \lambda \cdot \theta_2$ and $f_\lambda$ the ensembling of predictions $f_\lambda(\cdot) = (1 - \lambda) \cdot f(\cdot, \theta_1) + \lambda \cdot f(\cdot, \theta_2)$:*

$$f(\cdot, \theta_\lambda) \approx f_\lambda(\cdot)$$

*and for $k \in \{1, 2\}$:*

$$R_k(f(\cdot, \theta_\lambda)) \approx R_k(f_\lambda(\cdot))$$

*Proof.* This proof follows [68] and has two components.

**Functional approximation.** First, we perform a Taylor expansion at the first order of the models' predictions w.r.t. parameters $\theta$ for $x \in T$:

$$f(x, \theta_1) = f(x, \theta_\lambda) + \nabla_\theta f(x, \theta_\lambda)^\intercal (\theta_1 - \theta_\lambda) + \mathcal{O}\left(\|\theta_1 - \theta_\lambda\|_2^2\right)$$

$$= f(x, \theta_\lambda) + \nabla_\theta f(x, \theta_\lambda)^\intercal (\lambda \cdot \theta_1 - \lambda \cdot \theta_2) + \mathcal{O}\left(\|\theta_1 - \theta_2\|_2^2\right)$$

and similarly:

$$f(x, \theta_2) = f(x, \theta_\lambda) + \nabla_\theta f(x, \theta_\lambda)^\intercal ((\lambda - 1) \cdot \theta_1 + (1 - \lambda) \cdot \theta_2) + \mathcal{O}\left(\|\theta_1 - \theta_2\|_2^2\right)$$

Then by $\lambda$-weighted sum over $i$, the term multiplying $\nabla_\theta f(x, \theta_\lambda)^\intercal$ cancels out and we obtain:

$$f_\lambda(x) = (1 - \lambda) \cdot f(x, \theta_1) + \lambda \cdot f(x, \theta_2) = f(x, \theta_\lambda) + \mathcal{O}\left(\|\theta_1 - \theta_2\|_2^2\right). \tag{18}$$

**Reward approximation.** Second, we obtain the reward approximation with a Taylor expansion at the zeroth order of the reward $R_k$ for $k \in \{1, 2\}$ and injecting Equation (18):

$$R_k(f_\lambda(x)) = R_k(f(x, \theta_\lambda)(x)) + \mathcal{O}(\|f_\lambda(x) - f(x, \theta_\lambda)\|_2)$$

$$= R_k(f(x, \theta_\lambda)(x)) + \mathcal{O}\left(\|\theta_1 - \theta_2\|_2^2\right).$$

We obtain the results when $\theta_1$ and $\theta_2$ remain close, i.e., when we can ignore the $\mathcal{O}$ term. $\qquad\square$

## D  Text-to-text: LLaMA with diverse RLHFs

### D.1  Experimental details

We summarize the key implementation details of our text-to-text generation experiments in Table 1. The pre-trained network is LLaMA-7b [44]; then low-rank adapters [87] were fine-tuned on Alpaca [22] to follow instructions. We eventually fine-tune via PPO on the different considered tasks. Our code is adapted from [86]; we kept most of their hyperparameter values, only dividing by 2 the batch size to fit in our GPU and extending the output length. For each task, we consider existing open-source datasets[3] and available reward models, that we download from HuggingFace. Regarding the reward models, in summarization tasks, $R_1$ was open-sourced in an effort to reproduce the Summarize from Human Feedback paper [12], while $R_2$ [91] aimed at improved "faithfulness in abstractive summarization with contrast candidate generation". For other dialog tasks, we mostly rely on different reward models from OpenAssistant [92]; though they all aim at evaluating whether an answer is adequate given a question, they differ in their predictions due to differences in their architecture and training procedures. In practice, we leverage these reward models as block-box classification pipelines, implemented in the transformers library [82].

---

[3]For example, the TL;DR dataset is a previously existing dataset extracted and obtained by [89] that contains preprocessed comments posted on the social network Reddit and hosted on HuggingFace.

Table 1: LLaMA with RLHF experiments: key implementation details.

| Model | |
|---|---|
| Architecture | Transformer [75] |
| Pre-training | LLaMA-7b [44] |
| Instruction FT | Alpaca [22] |

| RL procedure | |
|---|---|
| Fine-tuning strategy | LoRA [87] |
| | *following Alpaca-LoRA [175]* |
| LoRA alpha | 16 |
| LoRA dropout | 0.05 |
| | *following trl-peft [85, 86]* |
| Optimizer | Adam [167] |
| Learning rate | 1.41e-5 |
| Batch size | 128 |
| Output length | Uniformly sampled between 16 and 32 |
| RL algorithm | PPO [84] |
| KL PPO | 0.05 for summary tasks else 0.2 |
| Epochs | 2 for Reuter summary else 1 |
| Hardware | NVIDIA RTX A6000 49 Go |
| Compute budget | 4000 GPUh |

| | **Reuter summary** |
|---|---|
| Task name | |
| Description | Generate a concise and clear summary of newspaper articles from Reuters. |
| Prompt | "Generate a one-sentence summary of this post." |
| Dataset | Reuter news from [88, 176] from news-summary |
| $R_1$ | gpt2-reward-summarization trained here. |
| $R_2$ | bart-faithful-summary-detector [91] |
| Figure | Figures 1(b) and 2(a) |

| | **Reddit TL;DR summary** |
|---|---|
| Task name | |
| Description | Generate a concise and clear summary of posts from Reddit across a variety of topics (subreddits). |
| Prompt | "Generate a one-sentence summary of this post." |
| Dataset | Reddit crawl from the TL;DR dataset [89] from summarize-from-feedback [12] |
| $R_1$ | gpt2-reward-summarization trained here. |
| $R_2$ | bart-faithful-summary-detector [91] |
| Figure | Figure 2(b) |

| | **Stack Exchange** |
|---|---|
| Task name | |
| Description | Answer accurately to technical questions from Stack Exchange. |
| Prompt | No prompt, only users' questions. |
| Dataset | Q&A from Stack Exchange [90, 177] from stack-exchange-preferences |
| $R_1$ | reward-model-deberta-v3-base |
| $R_2$ | reward-model-electra-large-discriminator |
| Figure | Figure 2(c) |

| | **Movie review** |
|---|---|
| Task name | |
| Description | Generate movie reviews that accurately describe a movie. |
| Prompt | "Generate a movie review." |
| Dataset | IMDB reviews [178] from IMDB |
| $R_1$ | reward-model-deberta-v3-base |
| $R_2$ | reward-model-electra-large-discriminator |
| Figure | Figure 2(d) |

| | **Helpful assistant** |
|---|---|
| Task name | |
| Description | Provide helpful and harmless answers to potentially complex and sensitive questions. |
| Prompt | No prompt, only users' questions. |
| Dataset | Helpfulness and harmlessness datasets [49] from hh-rlhf |
| $R_1$ | reward-model-deberta-v3-large-v2 |
| $R_2$ | reward-model-electra-large-discriminator |
| $R_3$ | reward-model-deberta-v3-base-v2 |
| $R_4$ | reward-model-deberta-v3-base |
| Figure | Figures 2(e) and 2(f) |

## D.2 Inspection of the predictions by the interpolated models

This supplementary material includes multiple visualizations of predictions generated by models with interpolated weights. For example, Table 2 provides generated summaries, that remain grammatically coherent. We include more generated samples on this website.

To quantitatively validate that samples generated by interpolated models do not suffer from reduced quality, we provide evaluations with general-purpose quality metrics. Specifically, in Figure 10, we measure the perplexity (of MLMS [179] or of GPT2 [180]) for summaries generated when $\lambda$-interpolating between two LLMs fine-tuned on the two summary rewards. We observe that, by interpolating the weights, we also interpolate the metrics. In conclusion, we confirm quantitatively that RS does not deteriorate quality.

Table 2: Examples of generated summaries about vampire vigilantes in Malawi news, for $(1 - \lambda) \cdot \theta_1 + \lambda \cdot \theta_2$ interpolating between $\theta_1$ and $\theta_2$ fine-tuned on $R_1$ (evaluating completeness) and $R_2$ (evaluating faithfulness).

| $\lambda$ | Generation |
|---|---|
| 0.0 | Malawi vigilante mobs have been killing people suspected of being vampires, prompting the UN and US embassy to declare no-go zones in the affected' |
| 0.1 | Malawi vigilante mobs have been killing people suspected of being vampires, prompting the UN and US embassy to declare no-go zones in the affected |
| 0.2 | Malawi vigilante mobs have killed nine people since mid-September, prompting the United Nations and the US embassy to declare some parts of the country no- |
| 0.3 | Malawi vigilante mobs have killed nine people, prompting the UN and US embassy to declare parts of the country no-go zones due to widespread |
| 0.4 | Malawi vigilante mobs have killed nine people, prompting the UN and US embassy to declare some parts of the country no-go zones. |
| 0.5 | Malawi vigilante mobs have arrested and killed suspected vampires, prompting the UN and US embassy to declare no-go zones and President Peter Muth |
| 0.6 | Malawi vigilante mobs have arrested and killed suspected vampires, prompting the UN and US embassy to declare no-go zones. |
| 0.7 | Malawi vigilante mobs have arrested suspected vampires, resulting in deaths and prompting the UN and US embassy to declare no-go zones. |
| 0.8 | Malawi vigilante mobs have arrested suspected vampires, resulting in deaths and violence. |
| 0.9 | Malawi vigilante violence has caused widespread panic and death, prompting authorities to arrest suspected members and investigate the belief in vampirism. |
| 1.0 | Malawi vigilante violence has caused widespread panic and death. |

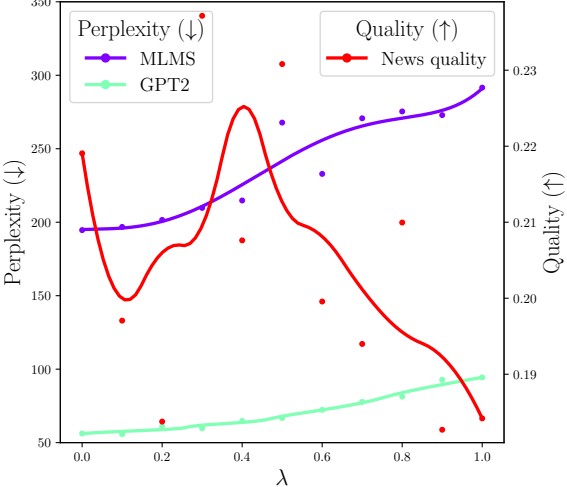

Figure 10: News summaries generated by $\lambda$-interpolated LLMs evaluated in terms of perplexity (by MLMS [179] or by GPT2 [180]) or news quality by this classifier.

# E  Image-to-text: captioning with diverse statistical rewards

## E.1  Experimental details

We summarize the key implementation details of our captioning experiments in Table 3. In short, we took the state-of-the-art network [96] for captioning on COCO, fine-tuned with their code and only changed the reward. In more details, since the *self-critical* paper [24] (a variant of REINFORCE [98] with a specific estimation of the baseline score) it is now common in captioning to optimize the CIDEr reward [31] after a first step of supervised fine-training. The recent ExpansionNetv2 [96] follows this strategy to reach state-of-the-art results, with a Swin Transformer [97] visual encoder and a block static expansion for efficiency. We investigate whether additional RL trainings on the other traditional statistical metrics can help. We use the code from [96] and their hyperparameters, only reducing the batch size from 24 to 18 to fit in our GPUs and consequently adapting the learning rate.

Table 3: Captioning experiments: key implementation details.

| Model | |
|---|---|
| Architecture | ExpansionNetv2 [96] |
| Visual encoder | Swin Transformer [97] |
| Visual encoder pre-training | ImageNet 22k [181] |
| Fine-tuning | Cross-entropy then CIDEr RL [24] on COCO [94] |
| **RL procedure** | |
| Fine-tuning strategy | Usually frozen visual backbone, but end-to-end in Figure 13(d) |
| RL algorithm | Self-critical [24], a variant of REINFORCE [98] |
| Optimizer | Radam [182] |
| Dataset | COCO [94] and Karpathy split [99] |
| Rewards | BLEU [29] (with 1-gram or 4-grams), ROUGE [30], METEOR [95], CIDEr [31] |
| Learning rate | 1e-5 |
| Batch size | 18 |
| Gradient accumulation | 2 |
| Warmup | Anneal 0.8 during 1 epoch |
| Epochs | 6 |
| Hardware | GPU V100 32G |
| Compute budget | 1500 GPUh |

## E.2  Additional results

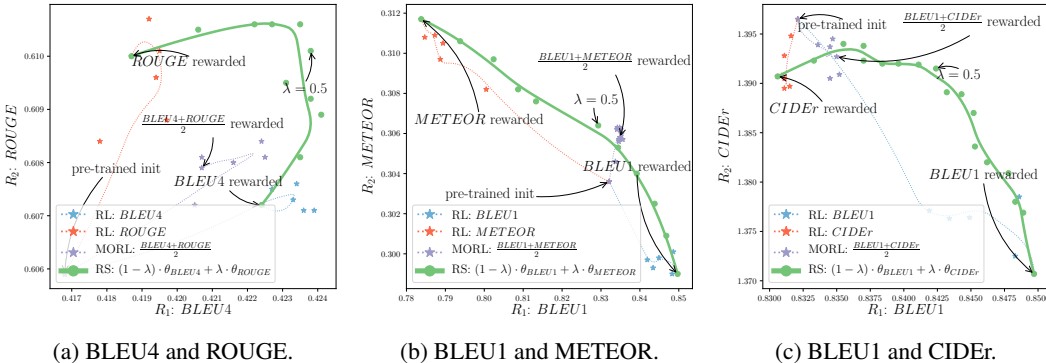

(a) BLEU4 and ROUGE.    (b) BLEU1 and METEOR.    (c) BLEU1 and CIDEr.

Figure 11: Additional results in captioning with more rewards, complementing Figure 3. Specifically, Figure 11(a) uses $R_1 = BLEU4$ and $R_2 = ROUGE$; then, with $R_1 = BLEU1$, Figure 11(b) uses $R_2 = METEOR$ and Figure 11(c) uses $R_2 = CIDEr$. In particular, the latter shows the failure when optimizing CIDEr; indeed, let's recall that the pre-trained initialization [96] has already been trained by optimizing CIDEr [24]. Thus optimizing CIDEr a second time does not help, neither in CIDEr nor in other rewards. That's why in Figure 3(c) we consider the initialization as the network parametrization optimized for CIDEr.

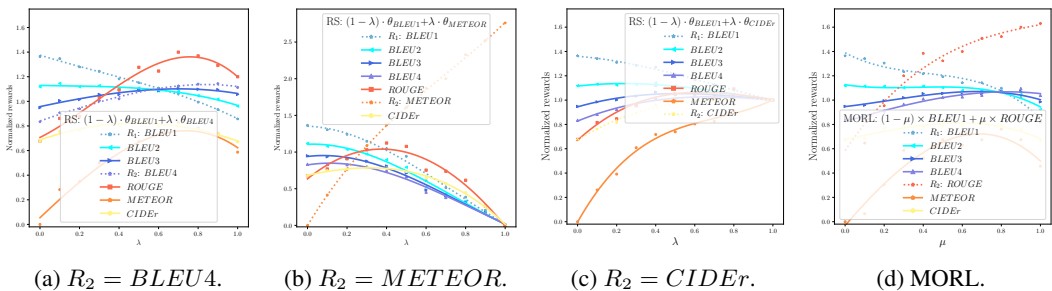

(a) $R_2 = BLEU4$. (b) $R_2 = METEOR$. (c) $R_2 = CIDEr$. (d) MORL.

Figure 12: Additional results in captioning when measuring performances on all rewards and varying the interpolating coefficients, complementing Figure 5(b). In Figures 12(a) to 12(c), we extend the results for RS with $R_1 = BLEU1$ and for varying $R_2$; the optimal $\lambda$ depends on the similarity between the evaluation metric and $R_1$ and $R_2$. We also see in Figure 12(c) that all rewards are normalized to 1 for the CIDEr-initialization. In Figure 12(d), we perform the same analysis for MORL while varying the weighting $\mu$ over the proxy rewards $R_1 = BLEU1$ and $R_2 = ROUGE$; we recover similar curves than in Figure 5(b) for RS.

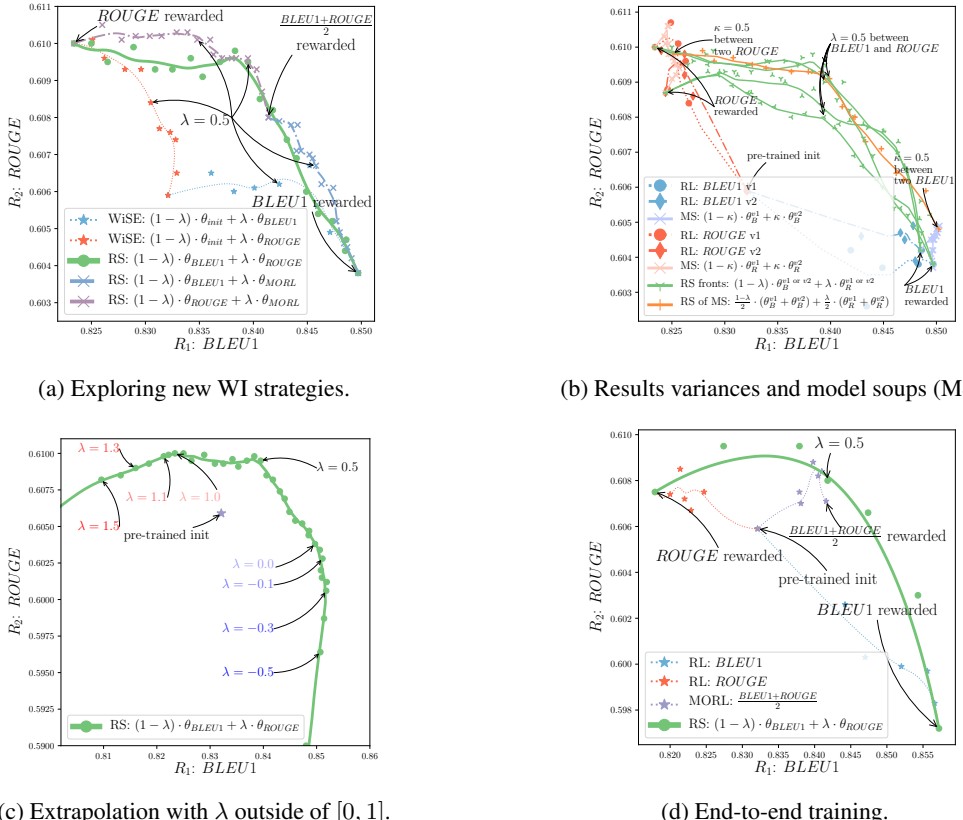

(a) Exploring new WI strategies.

(b) Results variances and model soups (MS).

(c) Extrapolation with $\lambda$ outside of $[0, 1]$.

(d) End-to-end training.

Figure 13: Additional results in captioning with $R_1 = BLEU1$ and $R_2 = ROUGE$. In Figure 13(a), we investigate interpolating the fine-tuned networks with the pre-trained initialization as in WiSE [183]; this only reveals a small portion of the front. In contrast, the interpolation with $\theta_{MORL}$ ($\mu = 0.5$) solution improves RS's front: this highlights some limitations in Hypothesis 2 and strict Pareto optimality of RS. Adding the MORL solutions as *intermediate* weights may help interpolate between two weights too distant. This suggests some practical complementarity between RS and MORL; given a training budget larger than the number of rewards, one may learn a few MORL for varying $0 \le \mu \le 1$, and then interpolate the obtained solutions. Figure 13(b) shows results' variance with two RL trainings for BLEU1, and two for ROUGE, each time with a different seed defining the data ordering and augmentations. Though we observe some randomness, the Hypothesis 1 is consistently validated. Moreover, it presents the fronts described when we interpolate weights fine-tuned on a shared reward, as in model soups (MS) [67, 68]; it mostly reduces variance and reveals only a small portion of the spectrum of preferences, validating the need to fine-tune on different rewards (as proposed in RS) to reveal the front across the entire space of preferences. Finally, the orange line shows that RS and MS can be complementary, by $\lambda$-interpolating the MS for BLEU1 and the MS for ROUGE with $\kappa = 0.5$. Figure 13(c) presents the extrapolation results when $\lambda$ goes outside of $[0, 1]$. This suggests that we can artificially reduce a reward with negative coefficients, as studied in [141]. Finally, Figure 13(d) shows the results when the networks are trained end-to-end, rather than keeping the backbone frozen. This validates the efficiency of rewarded soups in a new more general setting where all layers are trainable.

# F    Text-to-image: diffusion models with diverse RLHFs

## F.1    Experimental details

**Task description.** Several works have studied the problem of aligning the output of diffusion models with human feedbacks [25, 26, 33]. Notably, diffusion models can be fine-tuned to match human aesthetic perception. As for any subjective metric, there is a variety of reward models capturing different aesthetics. In our experiments, the two first reward models were trained in a supervised setting to match human quality ratings collected on large image datasets. Specifically, the first $R_1$ is the *ava* aesthetic model, available here, trained on 250.000 images from the AVA dataset [103], based on CLIP features. The second $R_2$ is the *cafe* aesthetic model, available here, trained on 3500 real-life and anime/manga images. Moreover, in Figure 14, we also consider a *nsfw* detector, estimating the probability of an image being *safe* by computing the cosine similarity with the CLIP embeddings of a set of *unsafe* words, as already done to filter the LAION dataset [184].

**Implementation details.** We use a 2.2B parameters diffusion model trained on an internal dataset of 300M images, which reaches similar generation quality as Stable Diffusion [102] in terms of CLIP alignment and FID scores on prompts from the 5000 images of the COCO test dataset (CLIPScore 30.0 vs 30.2 for Stable Diffusion, FID 19.0 vs 19.1 for Stable Diffusion). Given a reward model $R$, we first generate 10000 images with the pre-trained diffusion model on prompts from the COCO dataset, and compute the rewards for every generated image. For computational efficiency, we keep only a dataset $\mathcal{D}'$ containing the 50% images with the best scores, and rescale rewards $R$ linearly into $r$ so that $\min_{\mathbf{x}_0 \in \mathcal{D}'} r(x_0) = 0$ and $\frac{1}{|\mathcal{D}'|}\sum_{\mathbf{x}_0 \in \mathcal{D}'} r(x_0) = 1$. Then, we **fine-tune the diffusion model** on the reward-weighted negative log-likelihood [25]:

$$\mathcal{L} = \mathbb{E}_{(\mathbf{x}_0, Q) \in \mathcal{D}, \epsilon \sim \mathcal{N}(0,1), t \sim Uniform(0,T)} \quad r(\mathbf{x}_0) \times \|\epsilon_\theta(\mathbf{x}_t, t, Q) - \epsilon\|^2, \tag{19}$$

where $\epsilon_\theta$ is the noise estimation network, $T$ is the total number of training steps, $r(\mathbf{x}_0)$ is the rescaled reward of image $\mathbf{x}_0$ and $Q$ is the text associated to image $\mathbf{x}_0$. As a side note, on-policy RL would require performing loops of image generations and model fine-tunings [185], but we only perform a single *offline* iteration for simplicity. Moreover, for efficiency, we only fine-tune 10% of the diffusion model's weights [104] corresponding to the cross-attention layers and the bias/scaling parameters. As further described in Table 4, we apply the Adam [167] optimizer for 4000 steps with a batch size of 64 and a learning rate of 5e-6. To report results for each model (fine-tuned or interpolated via RS), we generate 1000 images from a held-out set of COCO prompts and then we average the scores given by the reward models. To reduce the variance in image generation, each prompt has a unique seed for all models, so that the input noise given to the diffusion model only depends on the text prompt.

Table 4: Image generation experiments: key implementation details.

| Model | |
|---|---|
| Architecture | GLIDE (2.2B parameters) |
| Pre-training | Internal dataset of 300M captioned images |
| **RL Procedure** | |
| Fine-tuning objective | Reward-weighted diffusion loss |
| Fine-tuned parameters | Cross-attention layers and bias/scale |
| Optimizer | Adam [167] |
| Dataset | Generated with COCO prompts |
| Rewards | *ava* [103] and *cafe* and *nsfw* |
| Learning rate | 5e-6 |
| Batch size | 64 |
| Epochs | 25 |
| Hardware | Single GPU V100 32G |
| Compute budget | 500 GPUh |

## F.2    Additional results

RS can trade-off between the two aesthetic rewards in Figure 6(a), allowing adaptation to the user's preferences at test time. Yet, we show some limitations in the spider map of Figure 14, when

computing MORL and RS on all three rewards: *ava*, *cafe* and also the *nsfw*. In this case, MORL has higher scores than RS. We speculate this is because the *nsfw* is very different from aesthetic preferences. Actually, the *nsfw* is inversely correlated with image quality: lower quality images result are less flagged as *unsafe*. This shows some limitations of weight interpolation when combining antagonist rewards. An improved strategy would first learn the MORL of the $N = 3$ rewards, and then optimize each reward independently from this improved initialization, before applying RS.

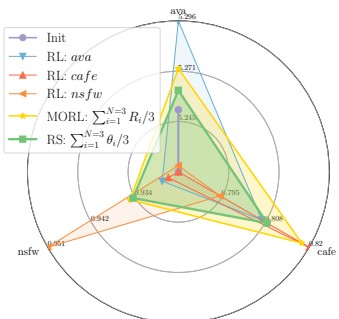

Figure 14: Image generation: spider map, with *ava*, *cafe* and *nsfw* reward models.

## F.3 Visualization of generated images from interpolated models

We show in Appendix F.3 images generated by rewarded soups when varying the interpolation coefficient $\lambda$ between the two models fine-tuned for the *ava* and the *cafe* aesthetic rewards. You can find additional qualitative results for this experiment on this website.

Moreover, in Appendix F.3, we measure the FID [186] and the CLIPScore [187] of the images generated by the same interpolated models. This confirms quantitatively that images generated by interpolated models remain coherent.

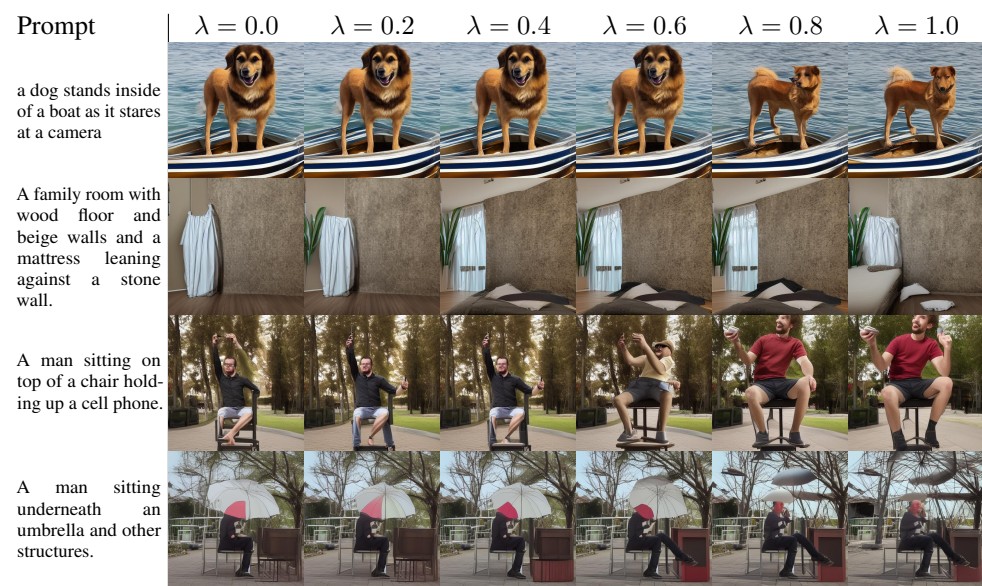

Figure 15: Visualization of images generated with rewarded soups for a varying interpolation coefficient $\lambda$ between the two models fine-tuned for the *ava* (corresponding to $\lambda = 0$) and *cafe* (corresponding to $\lambda = 1$) reward models. We can see that all interpolated models produce images of similar quality compared to fine-tuned models, demonstrating linear mode connectivity between the two fine-tuned models.

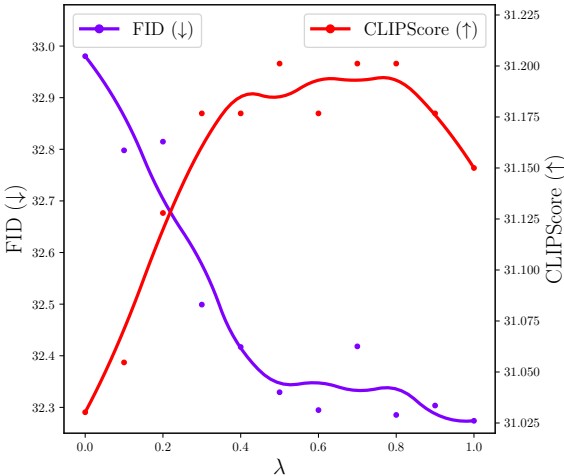

Figure 16: Images generated by $\lambda$-interpolated diffusion models evaluated in terms of realism by FID [186] or text alignment by CLIPScore [187].

# G   Text-to-box: visual grounding of objects with diverse sizes

## G.1   Experimental details

We show the implementation details in Table 5. We use UnIVAL [106], a model pre-trained solely on public benchmarks, to solve a variety of multimodal tasks such as VQA, visual grounding and image captioning. It is then fine-tuned on RefCOCO+ dataset for visual grounding. During the last fine-tuning phase, we complement the cross-entropy loss with an additional REINFORCE [98] term rewarding accuracy when the object is of the considered size. This means that the loss for $\theta_{Small}$ is $-\big(log(\hat{y}) + 5 \times 1_{\{\text{area}(\hat{y}) \text{ is small}\}} \times 1_{AUC(y,\hat{y})>0.5} \times log(y)\big)$ for an object with ground-truth box $\hat{y}$ and prediction $y$. The image is discretized into $1000 \times 1000$ bins before calculating the box areas. The task is illustrated in Figure 17.

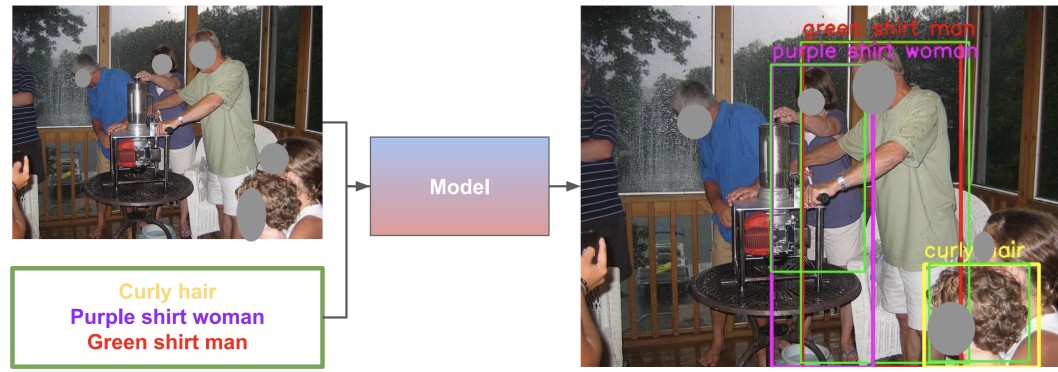

Figure 17: Illustration of the visual grounding task. The RS model results from the average of $N = 3$ weights specialized to detect respectively small, medium and large objects. The model takes a text (one description at a time) as input and outputs the bounding box in the corresponding region of the image. We show an example of small, medium and large predictions, and the associated ground truths in green. These texts and image are from the validation set of RefCOCO+ [105].

Table 5: Visual grounding experiments: key implementation details.

| Model | |
| --- | --- |
| Architecture | UnIVAL [106] |
| Visual encoder | ResNet-101 |
| Pre-training | Cross-Entropy on Public datasets (VQA, VG, Captioning) |
| Supervised fine-tuning | Cross-Entropy on RefCOCO+ [105] |
| **RL procedure** | |
| Fine-tuning strategy | end-to-end |
| Dataset | RefCOCO+ [105] |
| RL algorithm | Cross-entropy + $5\times$ REINFORCE |
| Reward Small | IoU>0.5 for object with area $< 30000$ |
| Reward Medium | IoU>0.5 for object with $30000 \leq$ area $< 100000$ |
| Reward Large | IoU>0.5 for object with $100000 \leq$ area |
| Optimizer | Adam |
| Learning rate | 3e-5 |
| Batch size | 256 |
| Epochs | 10 |
| Hardware | 8 GPU 60GB |
| Compute budget | 800 GPUh |

## G.2   Additional results

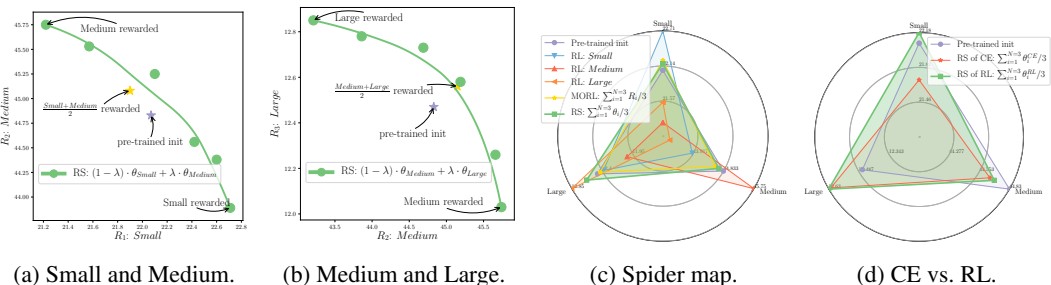

(a) Small and Medium.    (b) Medium and Large.    (c) Spider map.    (d) CE vs. RL.

Figure 18: Results in visual grounding on RefCOCO+ [105]. We use REINFORCE [98] to improve directly the non-differentiable accuracy, i.e., predict boxes with IoU> 0.5 w.r.t. the ground-truth. Fine-tunings are specialized on either small, medium, or large objects. These experiments complement Figures 6(b) and 18(c). Figure 18(c) shows that improving results on all sizes simultaneously is challenging, as MORL performs similarly to the initialization. Finally, Figure 18(d) motivates the use of RL to fine-tune on different sizes. Indeed, the results for (the proposed) RS of RL are significantly better than the results for RS of CE, where we average weights specialized on different sizes by fine-tuning with cross-entropy (rather than with REINFORCE).

## H   Locomotion with diverse engineered rewards

**Task description.** This experiment takes on the intricate challenge of controlling a running humanoid in the Brax [113] physics engine. The complexities involved in achieving natural or fast movement in continuous control environments serve as a testament to the robustness of our approach. The fine-tuning procedure is carried out on two distinct reward functions, with the aim of refining the running behavior of the humanoid, potentially resulting in smoother motion patterns. You can find qualitative results of this experiment on this website.

**Pre-training.** According to Remark 1, the LMC requires pre-training the base policy before fine-tuning. Thus, as the pre-training task, we use the default dense reward implemented in Brax: $R = velocity - 0.1 \times \sum_t a_t^2$. This pre-training phase also serves to collect statistics about observations and normalize them before inputting to the model (as it facilitates training). We used the Brax implementation of PPO [84]. The pre-trained policy is saved while the value function is discarded.

**Fine-tuning.** We keep the same environment as in pre-training. We also use the normalization procedure inherited from pre-training but freeze the statistics. Two reward functions are designed: a *risky* one for $R_1 = velocity$ and a *cautious* one where $R_2 = velocity - \sum_t a_t^2$. We tried a few hyperparameters (see the values in brackets in Table 6) but results (see Figure 19) remain close and consistently validate our working hypotheses.

Table 6: Locomotion experiments: key implementation details.

| **PPO Pre-training** | |
|---|---|
| Interactions | 5e8 |
| Reward Scaling | 1.0 |
| Episode Length | 1000 |
| Unroll Length | 10 |
| Discounting | 0.99 |
| Learning Rate | 5e-5 |
| Entropy Cost | 1e-3 |
| Number of environments in parallel | 4096 |
| Batch Size | 1024 |
| Hardware | 1GPU Tesla V100-SXM2-16GB |
| Runtime per experiment | 80min |
| **PPO Fine-tuning** | |
| Interactions | 1e8 |
| Reward Scaling | 1. |
| Normalize observations | True |
| Unroll Length | 10 |
| Discounting | {0.97, 0.99, 0.999} |
| Learning Rate | {1e-5, 3e-5, 1e-4} |
| Entropy Cost | {1e-3, 3e-3, 1e-2} |
| Number of environments in parallel | 4096 |
| Batch Size | 1024 |
| Hardware | 1GPU Tesla V100-SXM2-16GB |
| Runtime per experiment | 20min |
| **Model architecture** | |
| **Policy** | |
| Architecture | MLP |
| Nb of Layers | 6 |
| Hidden Size | 512 |
| **Value** | |
| Architecture | MLP |
| Nb of Layers | 5 |
| Hidden Size | 256 |

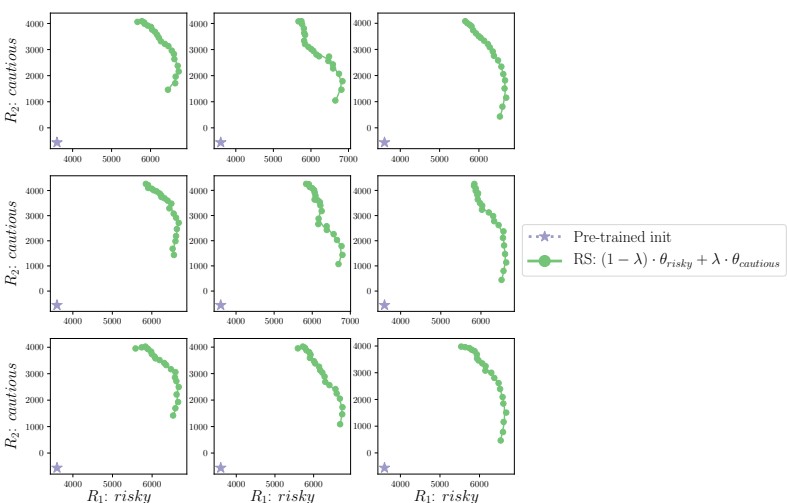

Figure 19: Analysis of results' variance for the locomotion task when varying the hyperparameters. Each column $i$ corresponds to the $i$-th $\theta_{risky}$, interpolated in case $(i, j)$ towards the $j$-th $\theta_{cautious}$. The Figure 6(c) is actually the plot from case $(1, 1)$.

