# Rewarded soups: towards Pareto-optimality by interpolating weights fine-tuned on diverse rewards

## Supplementary material

This supplementary material is organized as follows:

- Appendix A further discusses the practical benefits of rewarded soups.
- Appendix B details some theoretical guarantees.
- Appendix C details our text-to-text generation experiments.
- Appendix D enriches our image captioning experiments.
- Appendix E enriches our image generation experiments.
- Appendix F enriches our visual grounding experiments.
- Appendix G enriches our locomotion experiments.