# OpenReview forum: "Rewarded soups: towards Pareto-optimal alignment by interpolating weights fine-tuned on diverse rewards"
_NeurIPS.cc/2023/Conference — NeurIPS 2023 poster_

### Official Review · Reviewer_TSwH · 2023-07-06

**Soundness:** 4 excellent
**Presentation:** 4 excellent
**Contribution:** 3 good
**Rating:** 7
**Confidence:** 3

**Summary:**

The paper proposes rewarded soup (RS), a simple technique for combining policies trained for different rewards into a single policy performing well on a particular convex combination of those rewards. The technique consists in linearly interpolating weights of individual policies, using the fact that they share the same initialisation and remain linearly connected during finetuning on different rewards. The approach is well-motivated theoretically and thoroughly evaluated on a diverse array of tasks ranging from language generation, image captioning and image generation to robot locomotion.

**Strengths:**

1. The paper addresses crucial problems of accounting for diverse preferences and adapting to changes in reward specification that frequently arise in the emerging and important field of aligning generative models with human preferences.
2. Rewarded soup is well-motivated theoretically as a Pareto coverage set of policies for linear combinations of individual reward functions. I found working hypotheses 1 and 2 to be very helpful in understanding how RS works. I’m also convinced by empirical evidence for these hypotheses being true.
3. The fact that linear mode connectivity also holds for RL policies trained for different rewards is an interesting finding about deep learning overall. I found it somewhat surprising that interpolated weights outperform the interpolated rewards so consistently.
4. The paper is well-written and easy to follow despite being very dense. The theory part connects with experiments very well. I also appreciate the plots (e.g. Figure 2) being easy to navigate while conveying a lot of information.
5. The experiments are very thorough and diverse and I find them compelling.

**Weaknesses:**

I think the discussion of reward misspecification could be more nuanced. I think the claims that RS “mitigates reward misspecification” (line 75) and “If Hypothesis 2 is true, then RS can mitigate reward misspecification” (line 161) should be framed a bit more cautiously, making it clear that it’s a very particular kind of reward misspecification: when the real reward is linear in a set of proxy rewards. I don’t think this is representative of most kinds of reward misspecification that we see and that we should me worried about such as context-dependence of human preferences or biases of data workers providing feedbacks.

**Questions:**

1. What’s the relation between scatter points and curves on the plots: are curves a smoothed based on scatter points? What smoothing technique did you use?
2. How much does the performance of RS depend on policies being close to their shared base models? How does the RS front evolve over the course of finetuning; does it degenerate after some number of gradient updates?
3. Relatedly, does using parameter-efficient finetuning (e.g. LoRA) play a role? In the paper, we claim you use LoRA only for computational efficiency reasons, but shouldn’t it also significantly help to maintain linear mode connectivity? Does RS work without LoRA?

**Limitations:**

Limitations and societal impacts are discussed thoroughly.

---

> ### Author Rebuttal · Authors · 2023-08-04
>
> We would like to thank R.TSwH for this positive review and the great understanding of the empirical and theoretical components of our work.
>
> ---
>
> ### Q1. Reward misspecification
>
> In the revised version of the paper, we will clarify the discussion l.75 and l.161 on reward misspecification being mostly for linear rewards. Yet, please note that in Figure 4b (and in Figure 9 from Appendix D.2) we actually observe that "despite the lack of theoretical guarantees, weight interpolation improves results **even for non-linear reward**" (l.167). We speculate RS actually maximizes the projection of the user's reward on the linear subspaces defined by the different proxy rewards.
>
> ---
>
> ### Q2. Smoothing functions in plots
>
> The curves fit the points with a **Savitzky-Golay smoothing filter** (inspired from this [blog](https://www.datatechnotes.com/2022/05/smoothing-example-with-savitzky-golay.html)) and a **quadratic interpolation** (inspired from this [stack overflow](https://stackoverflow.com/questions/52014197/how-to-interpolate-a-2d-curve-in-python)). The code is detailed below.
>
> ```python
> import numpy as np
> from scipy.interpolate import interp1d
> from scipy.signal import savgol_filter
>
> def smoothing(x, y):
>     x_smooth = savgol_filter(x, 3, 1)
>     x_smooth[0], x_smooth[-1] = x[0], x[-1]
>     y_smooth = savgol_filter(y, 3, 1)
>     y_smooth[0], y_smooth[-1] = y[0], y[-1]
>
>     points = np.array([x_smooth, y_smooth]).T
>     distance = np.cumsum(np.sqrt(np.sum(np.diff(points, axis=0)**2, axis=1)))
>     distance = np.insert(distance, 0, 0) / distance[-1]
>     alpha = np.linspace(0, 1, 75)
>     interpolator = interp1d(distance, points, kind="quadratic", axis=0)
>     curve = interpolator(alpha)
>     return curve.T
> ```
>
> ---
>
> ### Q3. How much does the performance of RS depend on policies being close to their shared base models? How does the RS front evolve over the course of finetuning; does it degenerate after some number of gradient updates? (see [R.ntSF.Q1](https://openreview.net/forum?id=lSbbC2VyCu&noteId=UEM7DRMGYu))
>
> As detailed in Remark 1, "when the weights remain close, we can theoretically justify Hypotheses 1 and 2 (see Appendix B.2), and, more broadly, demonstrate that WI approximates ensembling (see Lemma 4 [in Appendix B.3])" (l.146). In other words, good performances are guaranteed when weights are close; thus longer trainings may be worrisome, as the models may potentially diverge in the weight space.
> We investigate this question in the one-page rebuttal pdf, for the news summarization task (in Figure 3.a) and for the captioning task (in Figure 3.b); we double the number of training steps, and report multiple RS fronts over the course of fine-tuning.
> Fortunately, we **consistently observe good performances for RS along the full fine-tuning**, confirming that the pre-trained initialization is sufficient to enforce the LMC, and validating the insights from previous works [Neyshabur2020] in supervised learning.
>
> [Neyshabur2020] What is being transferred in transfer learning? NeurIPS.
>
> ---
>
> ### Q4. Does RS work without LoRA?
>
> Actually, **most of our experiments are without LoRA**.
>
> - for the image generation task, we fine-tune 10% of the weights of the diffusion model, "corresponding to the cross-attention layers and the bias/scaling parameters" (l.1047).
> - for the visual grounding task, we fine-tune the transformer end-to-end.
> - for the locomotion task, we fine-tune the MLP end-to-end.
> - for the captioning task, we usually fine-tune the text decoder with the convolutional visual encoder frozen, but we show in Figure 10.d that RS convexity is actually even better when training end-to-end.
>
> Therefore, we argue that RS is agnostic to the "parameterization" strategy in training.
>
> As a final note, [Li2022] have observed in NLP that weight interpolation works even better in larger architectures. Intuitively, a large number of parameters may facilitate the orthogonality of the fine-tuned updates observed in [Ilharco2023], which "speculate that this [orthogonality] enables the combination of task vectors via addition with minimal interference". This fact may explain why end-to-end fine-tuning in captioning provides better convexity in Figure 10.d than when keeping the visual encoder frozen in Figure 3.a. Moreover, this insight suggests that, as LoRA reduces the number of trainable weights, **performances might actually get better with full end-to-end fine-tuning than with LoRA** (as currently done in our text-to-text experiments). This is a promising research direction for future work.
>
> [Li2022] Branch-Train-Merge: Embarrassingly parallel training of expert language models.\
> [Ilharco2023] Editing models with task arithmetic. ICLR.
>
> ---
> Thank you once more for your feedback. We remain open to further suggestions and discussions.

---

> > ### Comment · Reviewer_TSwH · 2023-08-15
> >
> > Thanks for the detailed response! I appreciate the experiment involving finetuning for more gradient steps. I stand by my (high) score.

---

### Official Review · Reviewer_bXWy · 2023-07-09

**Soundness:** 2 fair
**Presentation:** 3 good
**Contribution:** 2 fair
**Rating:** 4
**Confidence:** 4

**Summary:**

This paper presents reward soups (RS) which is the idea of starting with a pre-trained network, which is finetuned to multiple proxy rewards (say, multiple different criteria), and at test time, infers a reward as a linear combination of these proxy rewards and uses this to linearly combine the corresponding weights which is then used for prediction/generation. In contrast to naive variants of multi-objective RL which trains many different policies (far greater than the number of proxy rewards) to obtain a high fidelity policy for preferences encountered at test time, the proposed approach ends up working while training a number of policies that is equal to the number of proxy rewards while showing reasonable empirical performance.

==> post rebuttal: updated score.

**Strengths:**

The problem formation and proposed approach are topics of increasing interest and relevance to the community. The paper presents interesting results for many practically relevant and useful benchmarks.

**Weaknesses:**

- There are not much comparisons to approaches in multi-objective RL, which makes it unclear as to how one can imagine this paper's result to be approaching notions of pareto optimal trade-offs. While the paper acknowledges this issue, it leaves open huge gaps as to what can be achieved using a suite of approaches that exist in multi-objective RL (for e.g. even starting from reference 130 cited in this paper).

**Questions:**

Something that appears unclear is that linear mode connectivity etc. typically make sense in the context of supervised learning when training with the same set of labels. Here, the paper attempts to argue a similar perspective with different sets of labels, and in the RLHF context. Attempting to argue about both of these (particularly the latter) is non-trivial sense saying that a linear combination of parameters of a policy network optimizes the long term (i.e. generation level) linear combination of reward posits very strong assumptions on the structure of the optimal policy etc. for which I do not yet see a clear argument in this paper.

**Limitations:**

yes.

---

> ### Author Rebuttal · Authors · 2023-08-04
>
> We thank R.bXWy for reviewing our work. Yet, with all due respect, there is an inaccuracy in the summary by R.bXWy: "at test time, [we do **not**] infer a reward as a linear combination of these proxy rewards". More precisely, we show that interpolated weights can approximate the optimal policy for interpolated rewards, both empirically (in Section 3) and theoretically (in Appendix B.2), as detailed below.
>
> ---
>
> ### Q1. Towards Pareto-optimality and empirical comparisons with other MORL strategies
>
> As stated l.292, "when dealing with multiple objectives in deep learning, the common strategy is to combine them into a single reward"; in particular, the linear MORL is now standard to train LLMs with RLHF [Glaese2022]. Thus, "as the true Pareto front is unknown in real-world applications" (l.177), we use this linear MORL as the reference to evaluate Pareto-optimality. As explained in [R.QPfR.Q1](https://openreview.net/forum?id=lSbbC2VyCu&noteId=58jAIy2tXU), the key reason is that: "in full generality, improvements in initialization, RL algorithms, data, or specific hyperparameters could enhance performances. [Thus] the Pareto front [...] needs to be defined with regard to a training procedure" (l.151). Our conclusion was that RS is an empirical solution **towards** Pareto-optimality, with a limitation highlighted in the paper's name.
>
> Now, regarding the other MORL strategies, please note that **they are not practical** for large-scale experiments, as acknowledged by R.stHc who stated: "compared with previous work, [our approach is] much more applicable and flexible to complex application scenarios". For example, (i) "these works are mostly for **academic benchmarks**" (l.299) or "games such as ATARI" (l.890), and (ii) none have been used for RLHF, for fine-tuning foundation models, or for deep networks with billions of parameters.
> Critically, their implementations are complex, as most introduce **specific hyperparameters** or even "**modify the training procedure**" (l.300); for example, the reference 130 [Yang2019] requires a change in Bellman equations. In contrast, RS requires zero modification to the optimization algorithm (such as PPO), and thus can be used on top of any RLHF system (such as [TRL](https://github.com/lvwerra/trl)). If R.bXWy is aware of any open-source implementation of any MORL algorithm for RLHF of LLMs, we would run the experiments, report the numbers, but also verify if the LMC holds and **apply RS on top of those refined solutions**.
>
> Finally, **performance and simplicity are not the only advantages of RS over other MORL strategies, as discussed at length in Appendix A.2**. In brief, RS "is compatible with the inherent iterative engineering process of alignment" (l.890): "RS can continually include adjusted opinions while preventing forgetting of the old behaviours" (l.891). For example, if a new reward is defined, RS requires one single additional training, whereas the other MORL would require starting again from scratch.
>
> [Glaese2022] Improving alignment of dialogue agents via targeted human judgements.\
> [Yang2019] A generalized algorithm for multi objective reinforcement learning and policy adaptation. NeurIPS.
>
> ---
>
> ### Q2.  Linear mode connectivity and theoretical guarantees for the near-optimality of RS
>
> Our Hypothesis 1 tries to properly define the LMC when considering multiple metrics: R.TSwH "found [it] to be very helpful in understanding how RS works". Its empirical validation was arguably far from obvious; yet, we consistently obtain positive results in Section 3, for various setups and scenarios, even for generation task involving long term dependencies such as text-to-text with LLaMA, or image generation with diffusion models. Then, we state l.322: "RS relies on an empirical finding: the LMC, which currently lacks full theoretical guarantees [in our complex RL setup with multiple rewards, but actually] even in the simplest case of moving averages [Izmailov2018]" in supervised learning with one single set of labels.
>
> However, we'd like to respectfully emphasize that **we do provide a theoretical and novel "argument in this paper"**; in Appendix B.2 "we provide theoretical guarantees for the near-optimality of RS when considering quadratic rewards" (l.908). This is referenced in the main paper l.146 in Remark 1 and also l.141-143, where we state: "we theoretically prove in Appendix B.2 [that our Hypotheses 1 and 2] approximately hold when rewards are replaced by their second-order Taylor expansion with co-diagonalizable Hessians, a simplified setup justifiable when weights remain close", and a common assumption in deep learning (as argued in Remark 4). Specifically, considering a linear preference $\hat{\mu}$ over two rewards $R_1$ and $R_2$, Lemma 3 bounds the difference $\Delta R_{\hat{\mu}}$ between the rewards obtained by (i) the optimal policy and (ii) our interpolated solution by:
> $$ \Delta R_{\hat{\mu}} \leq \frac{\hat{\mu}^2(1-\hat{\mu})^2(M \Delta_1 - \Delta_2)(M \Delta_2 - \Delta_1)}{\left(\hat{\mu}(1-\hat{\mu})(M-1)^2 + M\right)\left(\left( 1 - \hat{\mu}\right) \Delta_1 + \hat{\mu} \Delta_2\right)},$$
> where $M$ is the maximum of eigenvalues ratio across the Hessians of the two rewards, $\Delta_1 = R_1(\theta_1) - R_1(\theta_2)$ and $\Delta_2 = R_2(\theta_2) - R_2(\theta_1)$. This bound is illustrated in Figure 7.
>
> In conclusion, we **provide guarantees with assumptions on the rewards** being quadratic and co-diagonalizable, thus with indirect "assumptions on the structure of the optimal policy" (R.bXWy). This is acknowledged by R.TSwH who stated: "the approach is well-motivated theoretically" and that "the theory part connects with experiments very well". This theoretical analysis will be put forward in the revision.
>
> [Izmailov2018] Averaging weights leads to wider optima and better generalization. UAI.
>
> ---
>
> If this clarifies our empirical and theoretical analyses, we would be extremely grateful if R.bXWy could update their review accordingly.

---

> > ### Comment · Reviewer_bXWy · 2023-08-18
> > **Response to author's rebuttal**
> >
> > Thank you for your response. I understand the relative merits offered by a simple strategy such as rewarded soups; while there are advantages to RS from the perspective of issues such as alignment, re-training, forgetting etc. that appear to be interesting, I was looking to understand what this method offered compared to an RL solution, and that hasn't been adequately addressed by the rebuttal.
> >
> > 1. I find notions of approximate optimality to not be well qualified without understanding what is achievable. This can be obtained through benchmarking some multi-objective RL methods which are directly related to the contributions of this paper. Since the method described in the paper is obviously simple, which is a positive, it will obviously serve the interest of the broader community to attempt to benchmark MORL methods from the literature. If this still is prohibitively infeasible as the authors appear to indicate, consider two possibilities:
> >
> > (a) I think running an RL method on a few grid points as a reward function and understanding what fraction of headroom is left out using this strategy of reward soups is necessary to round up on how good the proposed approach is. With regards to results that are presented in the paper in this front, why does running RL on the scalarized reward sometimes appear to be inferior to RS when RL can pretty much realize the RS solution?
> > (b) It will still be interesting to understand rewarded soups a bit more on locomotion tasks, where there is broad precedent of running multi-objective RL methods
> >
> >
> > Please note that RL is one of the central approaches that has helped improve alignment, and this disconnect with RL literature is a gap that I believe falls well within this paper's scope to be addressed. If the authors can address this comment (for instance, as they have presented results with (a)), I am open to improving my score.
> >
> > 2. Thanks for pointing to this. Again, can the authors clarify how sub-optimal the RS policy will be as a function of generation length and how this is captured by the bound?

---

> > > ### Author Response · Authors · 2023-08-18
> > >
> > > We would like to thank R.bXWy for acknowledging our rebuttal, and the merits offered by the simplicity of our strategy. The new comment suggests remaining concerns, that we try to clarify and answer below.
> > >
> > > ---
> > >
> > > ### "I find notions of approximate optimality to not be well qualified without understanding what is achievable. This can be obtained through benchmarking some MORL methods"
> > >
> > > Given a fixed preference $\hat{\mu}$ between two rewards $R_1$ and $R_2$, we would like to compare our RS policy to an oracle (but unavailable) policy maximizing $(1-\hat{\mu})R_1 + \hat{\mu} R_2$ in **test**. In practice, we believe that a sensible and a competitive approach is considering the model fine-tuned to maximize $(1-\hat{\mu})R_1 + \hat{\mu} R_2$ in **train**. This linearized MORL is then our reference to evaluate optimality.
> > >
> > > We want to clarify that the MORL references [124-133] from the related work Section 4 aim at efficiency, but (as far as we know) usually do not claim to consistently beat this linearized MORL.
> > > The only approaches that might improve performances are actually the references [168-171] from Appendix A.2, such as [Yu2020], that tackle gradient conflicts and different variance scales across tasks.
> > > Their contributions are orthogonal to our paper; yet, for the sake of completeness and to fill the gap between the RL and the alignment literature, we will include results for [Yu2020] on the locomotion task in the revision.
> > > If you think of any MORL method that might reveal a front better than RS's front, please tell us.
> > >
> > > [Yu2020] Gradient surgery for multi-task learning. NeurIPS.
> > >
> > > ---
> > >
> > > ### "running an RL method on a few grid points as a reward function and understanding what fraction of headroom is left out"
> > >
> > > We do something very similar when we quantitatively compare the hypervolume of RS to the hypervolume of the linearized MORL.
> > > Specifically, we take a few grid points with different interpolating coefficients ($\lambda$ for RS and $\mu$ for MORL), and then we compute their hypervolume, i.e., "the area over the curve wrt an optimal point" (l.198). This hypervolume helps measuring "what fraction of headroom is left out". We observe that "RS’s hypervolume is 0.367 vs. 0.340 for MORL in Figure 2(a), while it is 1.176 vs. 1.186 in Figure 2(b)" (l.199) in the summarization experiments, and that RS and MORL have the exact "same hypervolume of 0.140" (l.216) in the captioning experiment.
> > >
> > > ---
> > >
> > > ### "With regards to results that are presented in the paper in this front, why does running RL on the scalarized reward sometimes appear to be inferior to RS when RL can pretty much realize the RS solution?"
> > >
> > > Indeed, we observe a few times that the RS solutions are above the linearized MORL solutions. We speculate this is related to the multiple benefits of weight interpolation.
> > >
> > > - the main benefit that we discuss in our paper is the ability to interpolate between different policies. From this benefit, we would expect RS to perform similarly to MORL.
> > > - the second benefit from weight averaging is the implicit regularization, causing variance reduction and stabilizing performances. This is the main focus of the traditional weight averaging literature, for example in model soups. We speculate that this second benefit (combined with the first) can explain why RS sometimes outperforms MORL.
> > >
> > > ---
> > >
> > > ### How sub-optimal the RS policy will be as a function of generation length and how this is captured by the bound?
> > >
> > > We define l.82-84 "a policy by mapping inputs $x$ to $f(x, \theta)$ when parametrized by $\theta$. For a reward $\hat{R}$ [...] our goal is to maximize $\int_{x \in T} \hat{R}(f\left(x, \theta\right))$". In this setup, the $f$ includes the architecture choices, but also other design choices, such as (in our text experiments) the tokenization, the decoding strategy, and in particular the generation length.
> > >
> > > Then our Lemma 3 bounds the reward difference obtained for policies with (i) an optimal $\theta^*$ and (ii) RS interpolated weights.
> > > Critically, the rewards in the bound are obtained at fixed generation length for both policies.
> > > More generally, for fair comparison, our experiments are at fixed network and fixed training/inference procedures.
> > >
> > > Yet we acknowledge l.151 that "in full generality, improvements in initialization, RL algorithms, data, or specific hyperparameters could enhance performances". The key point is that RS could totally benefit from those improvements, for example longer generations.
> > > Actually, we validate this insight empirically for news summarization, when doubling the generation length at inference, from 32 to 64.
> > > We report below the scores for MORL with $\mu=0.5$ and for RS with $\lambda=0.5$.
> > > These results will be refined and included in the revision.
> > >
> > > |        | MORL  | MORL  | RS    | RS    |
> > > |--------|-------|-------|-------|-------|
> > > | Length | 32    | 64    | 32    | 64    |
> > > | $R_1$  | 1.31  | 1.27  | 1.45  | 1.47  |
> > > | $R_2$  | -1.00 | -0.95 | -1.11 | -1.03 |

---

> > > > ### Author Response · Authors · 2023-08-20
> > > >
> > > > As the discussion period ends soon, we want to notify that (so far) our implementation of [Yu2020] have not improved performances on the locomotion task. Once we have refined our results, we will include them in the revision. We will also test whether this gradient conflict MORL strategy can be applied on larger tasks.
> > > >
> > > > Finally, we thank R.bXWy and more generally all reviewers for the fruitful discussion. We hope they will consider our clarifications during discussions.

---

> > > > > ### Comment · Reviewer_bXWy · 2023-08-21
> > > > > **Re. author clarifications**
> > > > >
> > > > > Thank you to the authors for your clarifications. I still remain on the fence with the lack of understanding in terms of full fledged comparisons to multi-objective RL literature (there are no results available for reviewing purposes), and why running RL on the scalarized reward not being competitive with respect to RS. Note again, here, I don't think the author response of highlighting implicit regularization is convincing by itself and I think it is more than fair to believe that running RL with this linearized reward should at least be as good as RS (and more often than not, dominate RS) - this is because the RS solution is definitely realizable by running RL. This suggests there are some potential issues with running the RL related experiments that the authors need to look into more closely. All that being said, the RS results still appear to be pretty reasonable by themselves (leaving aside issues I mention above).
> > > > >
> > > > > I will increase my score, and will leave it to other reviewers to consider championing this paper.

---

> > > > > > ### Author Response · Authors · 2023-08-21
> > > > > > **Reproducibility**
> > > > > >
> > > > > > RS sometimes beats the linearized MORL because the weight interpolated solution is a regularized solution not *easily* realizable. Due to optimization challenges such as overfitting, the MORL may not generalize on test samples. In contrast, the regularization from weight averaging can help, in particular under train-test distribution shifts, as  analyzed in the out-of-distribution generalization literature: specifically, previous works showed that weight averaging leads to flatter hessians [94] and variance reduction [62,63,67,144].
> > > > > >
> > > > > > Regarding our experiments, as stated l.173, «for all tasks [and runs], we employ the default architecture, hyperparameters and RL algorithm; the only variation being the reward used across runs ». Thus the implementation for linearized MORL is literally weighting and summing different rewards. Actually, you can check out our released code, for example https://anonymous.4open.science/r/rewardedsoups-anonymous-6790/captioning/losses/reward.py for the captioning experiment where this phenomenon often occurs. You can also try to reproduce our results.
> > > > > >
> > > > > > As a final note, we would like to thank R.bXWy for the time and the engagement in the discussion.

---

### Official Review · Reviewer_QPfR · 2023-07-09

**Soundness:** 3 good
**Presentation:** 2 fair
**Contribution:** 2 fair
**Rating:** 6
**Confidence:** 3

**Summary:**

In this paper, the authors propose a multi-policy strategy called "rewarded soups" to fine-tune any foundation model, embracing the heterogeneity of diverse rewards. The method combines multiple networks through linear interpolation in the weight space, despite the non-linearities in the network, which efficiently yields Pareto-optimal solutions after training. The authors demonstrate the effectiveness of the approach for text-to-text, text-image, and locomotion control tasks, showing that "rewarded soups" can mitigate reward misspecification. The proposed approach aims to enhance the alignment of deep models and how they interact with the world in all its diversity. The authors highlight the issue of aligning AI systems to specific and diverse needs while making the process more transparent and limiting the cultural hegemony of a few individuals.

**Strengths:**

This paper addresses the reward misspecification problem caused by single priori rewards in current RLHF frameworks for foundation models. In order to solve the problem, this paper proposes a relatively complete framework called rewarded soup (RS). RS combines multiple networks (fine-tuning on different proxy rewards) through linear interpolation in the weight space and selects relative coefficients according to the user’s preferences, yielding Pareto-optimal solutions. The content of the whole paper is complete, and the experiments are sufficient.

**Weaknesses:**

1. The writing logic of the article could be more coherent in the reviewer's opinion. For example, in 3.3, "Moreover, RS gives a better front than MORL, validating Hypothesis 2." and in 3.5, "Moreover, the front defined by RS indicates an effective balance between risk-taking and cautiousness, providing empirical support for Hypothesis 2, although MORL with $\mu$ = 0.5 (i.e., $\alpha$ = 0.5) slightly surpasses RS's front."
2. Validations on Hypothesis 1 and 2 are based on empirical results. The authors are encouraged to give some theoretical analysis.
3. Lack of experiments on computational costs. According to the results, the proposed method has no strengths compared to MORL. The authors state that RS can reduce computational costs. However, no relative experiments showed in this paper.
4. Most experiments are assigned an N=2 reward model, which is not aligned enough with the "diverse" in the title.

**Questions:**

1. Why does the front passing through the point obtained by MORL fine-tuning on the average of the two rewards support Hypothesis 2 in Figure 2?
2. Some formulas need clarification. For example, is  $\lbrace\lambda_i\rbrace_{i}$ equals to $\lbrace\lambda_i\rbrace_{i=1}^N$?
3. $\lbrace\lambda_i\rbrace_{i=1}^N$ is selected by users according to their preferences. How do users select these coefficients? For example, the user will give a preference label over pair-wise (or k-wise) instances in the standard RLHF paradigm.

**Limitations:**

The authors have adequately addressed the limitations.

---

> ### Author Rebuttal · Authors · 2023-08-04
>
> We thank R.QPfR for reviewing our work, and try to address the expressed concerns below.
>
> ---
>
> ### Q1. Empirical validation of Hypothesis 2
>
> Our introduction in Section 3 and the Remark 2 explain why "the front passing through the point obtained by MORL fine-tuning on the average of the two rewards support Hyp. 2" (R.QPfR). This is because we use **this MORL as the reference to evaluate the Pareto-optimality of RS**. Specifically, we state l.176 that: "as the true Pareto front is unknown in real-world applications, we present empirical support for Hyp. 2 by comparing the front defined by RS (sliding $\lambda$ between $0$ and $1$) to the MORL's solutions optimizing the $\mu$-weighted rewards for $0\leq\mu\leq 1$ (sometimes only $\mu=0.5$ for computational reasons)". We provide more details l.151: "in full generality, improvements in initialization, RL algorithms, data, or specific hyperparameters could enhance performances. [Thus] the true Pareto front [...] needs to be defined with regard to a training procedure. [...] As such, in Section 3 [and Figure 2] we analyze Hyp. 2 by comparing the fronts obtained by RS and scalarized MORL".
>
> Then our experiments consistently show that MORL and RS perform similarly (with minor differences in different setups). Overall, in absence of the true Pareto front, this process provides empirical support **towards** Pareto-optimality of RS, with indeed a limitation highlighted in the paper's name.
>
> ---
>
> ### Q2. Theoretical analysis (extended in [R.bXWy.Q2](https://openreview.net/forum?id=lSbbC2VyCu&noteId=rSiwrlT8Be))
>
> Indeed, the *full validation* of "Hyp. 1 and 2 are based on empirical results" (R.QPfR). That"s why we state l.322: "RS relies on an empirical finding: the LMC, which currently lacks full theoretical guarantees, even in the simplest case of moving averages" in supervised learning.
>
> Yet, we respectfully disagree with R.QPfR as **our work already gives theoretical analysis**, in particular in Appendix B.2 where "we provide theoretical guarantees for the near-optimality of RS when considering quadratic rewards" (l.908). Specifically, in Lemma 3, we bound the reward difference between the optimal policy and our interpolated policy. This is referenced in the main paper at two different places, where we state: (i) l.141-143 "we theoretically prove in Appendix B.2 [that our Hypotheses 1 and 2] approximately hold when rewards are replaced by their second-order Taylor expansion with co-diagonalizable Hessians"; and (ii) l.146 "when the weights remain close, we can theoretically justify Hypotheses 1 and 2 (see Appendix B.2) and, more broadly, demonstrate that WI approximates ensembling (see Lemma 4)".
>
> ---
>
> ### Q3. Computational costs (close duplicate of [R.bJvT.Q3](https://openreview.net/forum?id=lSbbC2VyCu&noteId=Hztsf2jtoo))
>
> RS reduces the computational costs of other MORL strategies; eg, as stated in Figure 1.b, "with only two trainings [RS] reveals the green front of Pareto-optimal solutions [...] and matches the costly yellow front of MORL requiring [11] trainings on different linear weightings". As a side note, truly revealing the full MORL front would actually require an infinite number of trainings. Thus, we argue that this **efficiency gain is by design**; when considering $N$ rewards, RS only requires $M=N$ fine-tunings, while MORL "requires explicitly maintaining a large set $M\gg N$ networks, practically one for each possible preference" (l.105).
>
> To quantify the efficiency gain of RS, in Figure 2 (from the one-page rebuttal pdf) we define a new measure of success; the expected reward $E_{\hat{\mu}\sim Unif\left(0,1\right)} \hat{R_{\hat{\mu}}}$ where $\hat{R_{\hat{\mu}}} = (1-\hat{\mu})\times R_1 + \hat{\mu} \times R_2$ and the expectation is over all the possible user's linear preferences $\hat{\mu}$ over the $N=2$ rewards. Then we compute the difference between (i) the expected reward for RS (always with $2$ training runs), and (ii) the expected reward for MORL with $M$ training runs. **Plotting this expected reward advantage for different values of $M$ confirms that MORL needs $M \gg 2$ to match RS**. Moreover, because of the dimensional curse, we expect the number of MORL trainings required to match RS to grow exponentially with the number of rewards $N$. In conclusion, these new experiments quantitatively validate that RS is more efficient than MORL, and will be included in the revised paper.
>
> ---
>
> ### Q4. Number of rewards (extended in [R.ntSF.Q2](https://openreview.net/forum?id=lSbbC2VyCu&noteId=UEM7DRMGYu))
>
> For visualization clarity, the Pareto fronts were for $N=2$ rewards, one of the $x$-axis, the other on the $y$-axis. Yet, "RS can scale and trade-off between more rewards" (l.201). We validate this in the spider maps from Figure 2.f (for text generation), from Figure 3.c  (for image captioning, adapted in Figure 1.c and 1.d in the rebuttal), and from Figure 5.c (for visual grounding), where we respectively consider $4$, $5$ and $3$ different rewards.
>
> ---
>
> ### Q5. Formulas clarity
>
> Yes, both formulas refer to $\\{ \lambda_i \\}_{i=1}^N$. Bounds will be explicited in the revision.
>
> ---
>
> ### Q6. Selecting the $\lambda$
>
> As detailed l.163, and latter l. 223, we already **consider two practical strategies to select the values** of the interpolating coefficients $\lambda$:
>
> 1. if the user defines a linear preference $\hat{\mu}$, we can select $\lambda=\hat{\mu}$ .
> 2. if the user provides some labelled validation samples, we can cross-validate $\lambda$.
>
> We validate in Figure 4.a that both strategies perform well. If the user only provides preference comparisons (as suggested by R.QPfR), we could indeed select the $\lambda$ similarly as in reward modeling in RLHF.
>
> ---
>
> We would greatly appreciate it if you took those clarifications below into account during discussions.

---

### Official Review · Reviewer_bJvT · 2023-07-24

**Soundness:** 3 good
**Presentation:** 2 fair
**Contribution:** 2 fair
**Rating:** 4
**Confidence:** 3

**Summary:**

This paper explores a model-soup strategy to efficiently adapt to diverse reward functions from various real-world users. By fine-tuning a pre-trained LLM multiple times each with a specialized reward function and interpolating their weights linearly, the proposed method is able to adapt to various reward functions without having to train a new LLM per user.

**Strengths:**

- The proposed method is much more efficient than the baselines, which have to train a separate model when a new reward function is given.
- The evaluation is through and conducted on a variety of LLM tasks.
- The performance of the proposed method does not fall behind compared to the more costly baselines.

**Weaknesses:**

- The novelty of the paper is weak. It seems the main contribution of the paper is applying the weight interpolation (model soup [1]) technique, which was well-explored in supervised learning, to RLHF. I suggest the authors clarify the paper's novelty (other than applying the model-soup technique to another domain) more clearly.
- The authors point out that in RLHF, the reward function is different per each model, unlike supervised learning where the training objective is the same. I agree with the authors on this point, but the rewards used in the paper's experiments do not seem very heterogeneous to back up the authors' claim. A more realistic scenario would be to experiment with a set of rewards that contradict each other directly (e.g., different reward functions learned from users with conflicting interests).
- The main strength of the proposed method is that it is more efficient in terms of training (fine-tuning) cost and inference cost. However, the paper does not provide a quantitative comparison of these costs between the proposed method and the baselines. Therefore, it is hard to assess how much efficiency gain the proposed method will provide.

[1] Wortsman et al., Model soups: averaging weights of multiple fine-tuned models improves accuracy without increasing inference time, ICML 2022.

**Questions:**

Noted above.

**Limitations:**

Noted above.

---

> ### Author Rebuttal · Authors · 2023-08-03
>
> We thank R.bJvT for reviewing our work, and try to address the expressed concerns below.
>
> ---
>
> ### Q1. Novelty and difference with model soups (extended in [R.DNE5.Q1](https://openreview.net/forum?id=lSbbC2VyCu&noteId=6LMJAJD6vx))
>
> The first conceptual novelty is arguing for a **multi-objective paradigm** to align deep models with human preferences and reduce reward misspecification. The second empirical novelty is observing new setups/conditions where the **linear mode connectivity** holds, and thus where weight interpolation can be used; eg, in reinforcement learning with different rewards, for multimodal tasks. This weight interpolation strategy was indeed used in model soups (MS). Yet, we want to clarify that **RS and MS tackle different problems, have different goals, leading to different methods and implementations**.
>
> - MS challenges the standard model selection after a grid search to improve generalization in supervised learning, and aims at reducing the variance of the predictions by combining the fine-tuned models: thus MS considers different hyperparameters for a fixed training objective across runs, and (usually) uniform interpolating coefficients $\lambda=\frac{1}{M}$.
> - In contrast, RS challenges single-policy approaches to improve alignment in reinforcement learning, and aims at reducing reward misspecification by revealing a Pareto front of solutions across the entire space of preferences: thus RS considers different training objectives for fixed hyperparameters across runs, and non-uniform interpolating coefficients $\lambda$ set a posteriori.
>
> Overall, these differences mean that **RS can but MS cannot reduce reward misspecification**. We will clarify this difference between RS and MS in the revised version.
>
> ---
>
> ### Q2. Rewards diversity
>
> We respectfully disagree with R.bJvT, and argue that **we already use diverse and heterogeneous rewards that are in tension**. For example:
>
> - for the summarization tasks (in Figure 1.b, 2.a and 2.b): $R_1$ rewards completeness, while $R_2$ focuses on "faithfulness" (l.1005).
> - for the captioning experiments (in Figure 3.a and 8.b): BLEU1 measures accuracy while ROUGE captures recall (see l.42), and METEOR handles synonyms.
> - for the visual grounding experiments (in Figure 5.b and 14), the different rewards consider objects of different sizes.
>
> The dissimilarities between these rewards are quantitatively validated by our experiments; when fine-tuning on one reward, the performances are usually worsened on the others.
> For example, for captioning "tuning solely BLEU1 sacrifices some points on ROUGE" (l.213); for visual grounding "optimizing for small objects degrades performance on large ones" (l.259).
> These examples are arguably representative of "different reward functions learned from users with conflicting interests" (R.bJvT).
>
> Yet, we acknowledge (in Appendix E.2 and in our response to [R.ntSF.Q1](https://openreview.net/forum?id=lSbbC2VyCu&noteId=UEM7DRMGYu)) some "limitations of weight interpolation when combining antagonist rewards" (l.1059). This was suggested by the results for text-to-image generation in Figure 10, where RS underperforms MORL when considering a *nsfw* reward "very different from aesthetic preferences" (l.1057) and "inversely correlated with image quality" (l.1058). This limitation will be clarified in the limitation section from the revision. However, we want to emphasize that, in this kind of situation with fully antagonist rewards, the **complementarity of MORL and RS is a promising research direction**, as discussed l.1060: "an improved strategy would first learn the MORL [...], and then optimize each reward independently from this improved [MORL] initialization, before applying RS". As another research direction, we suggest (in the legend from Figure 10.a) that: "adding the MORL solutions as intermediate weights may help interpolate between two weights too distant".
>
> ---
>
> ### Q3. Quantify the efficiency gain (close duplicate of [R.QPfR.Q3](https://openreview.net/forum?id=lSbbC2VyCu&noteId=58jAIy2tXU))
>
> Indeed, "the main strength of the proposed method is that it is more efficient in terms of training (fine-tuning) cost" (R.bJvT) than the MORL baseline. For example, as stated in the legend from Figure 1.b, "with only two trainings [RS] reveals the green front of Pareto-optimal solutions [...] and matches the costly yellow front of MORL requiring [11] trainings on different linear weightings".
> As a side note, truly revealing the full MORL front would actually require an infinite number of trainings.
>
> Therefore, we argue that this **efficiency gain is by design**; when considering $N$ rewards, RS only requires $M=N$ fine-tunings, while MORL "requires explicitly maintaining a large set $M \gg N$ networks, practically one for each possible preference" (l.105).
> Indeed, as stated l.106, a critical issue in MORL is that “minor [preference] variations may result in significant changes in the solution. Thus, a high level of granularity in the mesh is necessary".
>
> To quantify the efficiency gain of RS, we now provide an analysis in Figure 2 from the one-page rebuttal pdf, where we define a new measure of success; the expected reward $E_{\hat{\mu}\sim Unif\left(0,1\right)} \hat{R_{\hat{\mu}}}$ where $\hat{R_{\hat{\mu}}} = (1-\hat{\mu})\times R_1 + \hat{\mu} \times R_2$ and the expectation is over the possible user's preferences $\hat{\mu}$. Then we compute the difference between (i) the expected reward for RS (always with $2$ training runs), and (ii) the expected reward for MORL with $M$ training runs. **Plotting this expected reward advantage for different values of $M$ confirms that MORL needs $M \gg 2$ to match RS**. Moreover, because of the dimensional curse, we expect the number of MORL trainings required to match RS to grow exponentially with the number of rewards $N$. In conclusion, these new experiments quantitatively validate that RS is more efficient than MORL, and will be included in the revised paper.

---

> > ### Author Response · Authors · 2023-08-21
> >
> > Dear Reviewer,
> > before the discussion period ends, we would love to know if you had the time to read our rebuttal, and whether additional clarification is required. Thank you again for reviewing our work.
> > Authors.

---

### Official Review · Reviewer_ntSF · 2023-07-24

**Soundness:** 3 good
**Presentation:** 3 good
**Contribution:** 3 good
**Rating:** 6
**Confidence:** 3

**Summary:**

The authors present a new strategy to address the heterogeneity of diverse rewards in reinforcement learning. Specifically, they propose 'rewarded soup,' which involves individually training multiple networks, each assigned to a different proxy reward, and then linearly combining these networks. Compared to the multi-objective reinforcement learning baselines, the proposed rewarded soup demonstrates its superiority on several benchmarks, including text-to-text, text-to-image, and control benchmarks.

**Strengths:**

- The authors presented a comprehensive study on the topic and the research field, as most arguments in introduction are supported by some references. The motivation is strongly supported and the path to the proposed method is reasonable.

- Presentation is clear, concise, and easy-to-understand, and the idea is simple yet effective.

- The authors conducted extensive experiments to verify the effectiveness of the proposed rewarded soup, and this includes multiple text-to-text tasks (shown in Section 3.1), image-to-text tasks (Section 3.2), text-to-image tasks (Section 3.3), and control tasks (Section 3.5). For most of the experiments, the improvement against MORL is obvious.

**Weaknesses:**

Although there are many strengths in the paper, there is a weakness that can be further enhance the overall quality.

- Ablation studies could be added: While the authors presented many reinforcement learning applications and also showed the improvement, it would be better to include more fine-grained ablation studies, such as how the difference of rewards affects the effectiveness of MORL baseline and the rewarded soup or how the number of networks affects the results.


**Questions:**

- This is aligned with the weakness part. I am curious about the effectiveness of the rewarded soup under different scenarios:

1. How does the difference/gap of rewards affect the effectiveness of the MORL baseline and the rewarded soup?
2. How does the number of networks affect the results?

The reason why I am particularly interested in these two questions is that a prior work [1] indicates when the models are quite different (based on their objectives), the linear combination is likely to produce less favorable results. I am wondering about "the limit" of the rewarded soup and in what situations the proposed method might fail.


[1] "Robust fine-tuning of zero-shot models" Wortsman et al., CVPR 2022

**Limitations:**

The authors have provided sufficient information of limitations and societal impact in the paper.

---

> ### Author Rebuttal · Authors · 2023-08-03
>
> We thank R.ntSF for the deep understanding of the paper, for highlighting its strengths and for asking two intriguing questions - that we try to answer below.
>
> ---
>
> ### Q1. How does the difference/gap of rewards affect the effectiveness of the MORL baseline and the rewarded soup?
>
> Our experiments in captioning and image generation provide empirical evidence that **the more similar the rewards, the higher the gains of RS versus MORL**.
>
> First, in the **captioning** experiment from in Figure 3.c, by analyzing the transfer abilities across the 4 main metrics (BLEU1, BLEU4, ROUGE, and METEOR), we can deduce that:
>
> - BLEU4 and ROUGE are very similar.
> - BLEU1 and BLEU4 are more similar than BLEU1 and ROUGE.
> - METEOR is an outlier, quite different from other metrics, in particular from BLEU1.
>
> Now having set these similarities across rewards, we observe that the gains of RS versus MORL are consistent with these similarities across rewards.
> Specifically,
>
> - with $R_1=\text{BLEU4}$ and $R_2=\text{ROUGE}$, we observe large performance gains for RS versus MORL (in Figure 8.a), where the green front is highly convex far above the solution provided by the MORL objective.
> - with $R_1=\text{BLEU1}$, we observe larger gains (and cleaner convexity) for RS versus MORL with $R_2=\text{BLEU4}$ (in Figure 3.b) than with $R_2=\text{ROUGE}$ (in Figure 3.a).
> - with $R_1=\text{BLEU1}$ and $R_2=\text{METEOR}$, we observe better performances for MORL than for RS (in Figure 8.b).
>
> Overall all captioning rewards remain sufficiently similar to favor RS over MORL when combining all rewards in Figure 3.c.
>
> Similarly, in the **image generation** experiment, when we consider two (arguably similar) aesthetic rewards in Figure 5.a to fine-tune a diffusion model, RS's front is to the right and above MORL's front.
> Then, when the rewards are very different or antagonist, we totally agree with your statement that "the linear combination is likely to produce less favorable results": in Appendix R.2 we include a *nsfw* reward "inversely correlated with image quality" (l.1058), and observe in this case that "MORL has higher scores than RS" (l.1056). This result "shows some limitations of weight interpolation when combining antagonist rewards" (l.1059). These insights were already briefly mentioned in the main paper (we state l.140 that "we report a few limitations in Appendix and research directions to fix them"), but will be clarified in the limitation section from the revised version. They can be explained in two different ways:
>
> - intuitively, from a **loss landscape perspective**, weights fine-tuned on diverse rewards will be more distant, thus potentially breaking the linear mode connectivity.
> - theoretically, thanks to **Lemma 3 in Appendix B.2.2**, where we bound the difference between the optimal reward and RS's reward by a RHS term growing with "the maximum of eigenvalues ratio" (l.942) across rewards' Hessians. This RHS term is illustrated in Figure 7. Then, if the rewards are more diverse, their Hessians would have more different eigenvalues, thus maximum of Hessian's eigenvalues ratio would grow, the RHS term would grow in Lemma 3, and our guarantees for the optimality of RS would get loose.
>
> ---
>
> ### Q2. How does the number of networks affect the results?
>
> Though most of our experiments are with $N=2$ rewards and networks for visualization clarity, "RS can scale and trade-off between more rewards" (l.201).
> We validate this empirically in the spider maps from Figure 3.c (for image captioning), from Figure 2.f (for text generation),, and from Figure 5.c (for visual grounding), where we uniformly combine respectively $M=5$, $M=4$, and $M=3$ networks fine-tuned on $N=M$ rewards, one reward each.
>
> Most importantly, we **refine this analysis** for the captioning task through additional spider maps in Figure 1.c and 1.d (from the one-page rebuttal). Specifically, we compare the performances across all $N=5$ metrics when averaging $1\leq M \leq 5$ networks (each fine-tuned on one of the $N$ rewards, thus leaving out $N-M$ rewards at training) and sequentially adding more networks to the weight average. We consistently observe that **adding one additional network specialized on one additional reward extends the scope of the possible rewards that RS can tackle Pareto-optimally**.
>
> As a short note, another possibility to scale the number of networks is at fixed number of rewards, by learning multiple networks on the same reward as in model soups (MS) [Wortsman2022].
> We consider this in the Figure 3.c from the one page rebuttal for the captioning task and plot $\lambda \to \frac{1-\lambda}{2} \cdot (\theta_{BLEU1}^{v1} + \theta_{BLEU1}^{v2}) + \frac{\lambda}{2} \cdot (\theta_{ROUGE}^{v1} + \theta_{ROUGE}^{v2})$, where $\theta_{BLEU1}^{v1}$ and $\theta_{BLEU1}^{v2}$ are from two independent RL fine-tunings on BLEU1 (and similarly for ROUGE). This shows that we can combine RS and MS, i.e., $\lambda$-interpolating between two MS weights specialized on different rewards, which are themselves the averages of models fine-tuned on the same reward.
>
> In conclusion, in all our experiments, **performances consistently increase for more networks**; when they are trained on different rewards, this reduces reward misspecification; when they are fine-tuned on the same reward, this reduces variance. This is consistent with the findings from previous works, eg, the Figure B.1 from model soups [Wortsman2022] which showed than increasing the number of averaged models consistently helps.
>
> [Wortsman2022] Model soups: averaging weights of multiple fine-tuned models improves accuracy without increasing inference time. ICML.
>
> ---
>
> We hope these explanations and these additional ablations clarify your questions. Please let us know if there is anything else we can do to further strengthen our submission.

---

> > ### Comment · Reviewer_ntSF · 2023-08-15
> >
> > Thanks authors for providing the response, and they are convincing and informative. I think these results support the arguments and properly answer my questions. I will keep my score as weak accept.

---

### Official Review · Reviewer_DNE5 · 2023-07-26

**Soundness:** 3 good
**Presentation:** 3 good
**Contribution:** 3 good
**Rating:** 5
**Confidence:** 4

**Summary:**

This manuscript studies a way to interpolate trained networks' parameters for diverse rewards in a reinforcement learning manner. To be specific, the proposed method introduces a way to achieve Pareto-optimal solutions through linearly weighted parameters after training. Extensive experiments showed the effectiveness of the proposed method on various domains such as text2text, image2text, and text2image.


**Strengths:**

- The manuscript is well-written and organized overall.
- The proposed idea is effective yet efficient as it does not require additional training.
- Extensive experimental results demonstrate the effectiveness of the proposed method on various domains; text generation, image captioning, and diffusion model.

**Weaknesses:**

Interpolating weights for better performance is not a new concept; model soups, which is mentioned in the manuscript. However, the authors did not provide any comparison with it. When we have N fine-tuned models, rewarded soup can perform better than model soup?
For example, in the case of an image captioning task, the experimental setup assumes only two rewarded models, differently fined-tuned models on AVA and cafe datasets, respectively. I think there is no reason to hesitate to apply a way of model soup.


**Questions:**

I already mentioned above in the weakness section. I would like to ask the authors how the proposed method is significantly better than other weight interpolation methods like model soup, not only the reward interpolation method, MoRL.

**Limitations:**

I think there is no potential negative societal impact.

---

> ### Author Rebuttal · Authors · 2023-08-03
>
> We thank R.DNE5 for highlighting the organization of the paper and the diversity of our experiments.
>
> ---
>
> ### Q1. Similarity and differences with model soups
>
> R.DNE5's main concern relates to the similarity between rewarded soups (RS) and model soups (MS).
> First, we totally acknowledge similarity with MS; actually, as stated l.65, "the name rewarded soups follows the terminology of model soups".
> Indeed, RS and MS both average the weights of models fine-tuned from a shared pre-trained initialization.
>
> Yet, we want to clarify that **RS and MS tackle different problems, have different goals, leading to different methods and implementations**.
>
> - MS challenges the standard model selection after a grid search to improve generalization in supervised learning, and aims at reducing the variance of the predictions by combining the fine-tuned models: thus MS considers different hyperparameters for a fixed training objective across runs, and (usually) uniform interpolating coefficients $\lambda=\frac{1}{M}$.
> - In contrast, RS challenges single-policy approaches to improve alignment in reinforcement learning, and aims at reducing reward misspecification by revealing a Pareto front of solutions across the entire space of preferences: thus RS considers different training objectives for fixed hyperparameters across runs, and non-uniform interpolating coefficients $\lambda$ set a posteriori.
>
> Overall, these differences mean that **RS can but MS cannot be applied to reduce reward misspecification**.
> We refer R.DNE5 to the Figure 10.b from Appendix D.2 (reproduced and enriched in the Figure 3.c from the one page rebuttal). The experiments are for the captioning task, when considering BLEU1 and ROUGE as rewards. The green lines only consider one fine-tuning per reward (standard RS), while  the light-blue (for BLEU1) and pink (for ROUGE) lines consider two fine-tunings on one single reward (standard MS).  As stated in the legend from Figure 10.b , "it presents the fronts described when we interpolate weights fine-tuned on a shared reward, as in model soups. This also only reveals a small portion of the spectrum of preferences, validating the need of diverse rewards to satisfy all users’ preferences". Specifically, MS mostly reduces variance; in contrast, considering **weights specialized on different rewards** (as proposed in this work) is key to reveal the front across the entire space of preferences.
>
> In summary, regarding the exact statements from R.DNE5:
>
> - "Interpolating weights for better performance is not a new concept": indeed, but we are the first to use weight interpolation for alignment, for models RL fine-tuned with different rewards, in particular for generative and multimodal tasks.
> - "the authors did not provide any comparison with model soups": actually we already did, in Figure 10.b from Appendix D.2.
> - "When we have $N$ fine-tuned models, rewarded soup can perform better than model soup?": RS will be better in terms of Pareto optimality. Yet, if the true reward is available before training and thus there is no reward misspecification, fine-tuning $N$ models on this exact reward (as in MS) will certainly provide better results.
> - "For example, in the case of an image captioning task, the experimental setup assumes only two rewarded models, differently fined-tuned models on AVA and cafe datasets". In the image captioning task (Section 3.2), we consider multiple metrics such as BLEU1, BLEU4, ROUGE, and METEOR. In the image generation task (Section 3.3), we consider two models fine-tuned on reward models trained on AVA and cafe datasets. In the latter case, fine-tuning multiple times on the cafe reward would fail to improve the AVA reward, as "the model $\theta_{\text{cafe}}$ performs poorly in terms of AVA" (l.245).
> - "how the proposed method is significantly better than other weight interpolation methods like model soup": RS is the only weight-interpolation method seeking Pareto optimality across diverse rewards, thus the other methods will only optimize a metric given a priori, without tackling reward misspecification.
>
> As a final note, in the Figure 3.c from the one page rebuttal, we also try to combine RS and MS, and thus plot:
> $\lambda \to \frac{1-\lambda}{2} \cdot (\theta_{BLEU1}^{v1} + \theta_{BLEU1}^{v2}) + \frac{\lambda}{2} \cdot (\theta_{ROUGE}^{v1} + \theta_{ROUGE}^{v2}),$ where $\theta_{BLEU1}^{v1}$ and $\theta_{BLEU1}^{v2}$ are from two independent RL fine-tunings on BLEU1 (and similarly for ROUGE). This orange line $\lambda$-interpolates between two MS weights specialized on different rewards, which are themselves the averages of models fine-tuned on the same reward. The convexity of the interpolation and the slightly better performances at the endpoints  show that **we can combine the benefits from RS (reward misspecification) and MS (variance reduction)**.
>
> ---
>
> We hope this answer clarifies the difference between rewarded soups and model soups; we remain available for any further discussion.

---

> > ### Comment · Reviewer_DNE5 · 2023-08-19
> >
> > I appreciate your response. I have read the other reviews that have been posted and their corresponding author responses. While some of my concerns have been addressed, it remains uncertain whether all the concerns of the other reviewers have been resolved. As of now, I maintain my current score for this paper, but I will continue to read other reviews and the authors' responses until the end of the discussion period.

---

> > > ### Author Response · Authors · 2023-08-19
> > >
> > > We thank R.DNE5 for taking the time to consider and acknowledge our rebuttal. Should there be any remaining concerns that you believe warrant further attention, we would be more than happy to provide additional clarification.

---

### Official Review · Reviewer_stHc · 2023-07-28

**Soundness:** 3 good
**Presentation:** 3 good
**Contribution:** 3 good
**Rating:** 5
**Confidence:** 3

**Summary:**

This paper proposes a method of using linear interpolated weight finetuned on different rewards instead of using linear combination of rewards to finetune weight, which is a solution to applying model under different and multiple preference scenarios. The idea is intuitive but works well, it (the RS) can achieve similar performance compared with MORL, while RS reduces the computational cost significantly. The author makes some mathematical hypotheses which empirically hold when all the weight to be interpolated is finetuned from a pretrained model. Also, the author has done a lot of experiments to show the feasibility of the proposed method.

**Strengths:**

The paper is clearly written. The experimental work is sufficiently done.

The author did a lot of experiments on different task which shows that this interpolating strategy is universal under different application scenarios, while with good performance.

The proposed method reduces the heavy computational requirements for pretuning compared with previous work, which makes it much more applicable and flexible to complex application scenarios.


**Weaknesses:**

Novelty: The most heavy workload in this paper is applying the strategy to different tasks and testing their performance, while less novel concepts or theories is presented.

Condition for Hypothesis: The hypothesis used in the method, such as the linear mode connectivity which states that the combined reward is concave to the weights requires research on the model structure and activation function. There should be some limitations for the design of networks which ensures that the hypothesis holds.

**Questions:**

Does the method harm the absolute quality of the produced samples? Evaluation other than reward functions should be provided.

May be you can show some extrapolated examples generated through this method.

**Limitations:**

All right in total.

---

> ### Author Rebuttal · Authors · 2023-08-03
>
> We thank R.stHc for the positive feedback on the clarity of our idea and the experiments. We would like to respond to R.stHc's review as follows.
>
> ---
>
> ### Q1. Novelty
>
> Our approach is novel from two perspectives.
>
> The first **conceptual** novelty is arguing for Pareto-optimality and a **multi-objective paradigm** to reduce **reward misspecification** when aligning deep generative models. This first novelty is critical to "handle the diversity of human preferences" (l.50), and as further detailed in Appendix A.1, to "support decision-making" (l.872), "interpretability and explainability" (l.878).
>
> The second **empirical** novelty is proposing rewarded soups, based on new setups/conditions where the **linear mode connectivity** holds (in reinforcement learning, with diverse rewards, even in the multimodal case) and thus where weight interpolation can be used. This second novelty is critical to reduce "the computational, memory, and engineering costs involved" (l.106) in traditional MORL approaches, and as further detailed in Appendix A.2, to be "compatible with the inherent iterative engineering process of alignment" (l.890).
>
> Moreover, we want to point out that in Appendix B.2 "we provide **theoretical** guarantees for the **near-optimality of RS** when considering quadratic rewards" (l.908), as referenced l.141-143 and l.146 in Remark 1. Specifically, in Lemma 3, we bound the reward difference between the optimal policy and our interpolated policy. We give more theoretical details in our response to [R.bXWy.Q2](https://openreview.net/forum?id=lSbbC2VyCu&noteId=rSiwrlT8Be).
>
> ---
>
> ### Q2. Limitations for the design of networks for LMC?
>
> In our experiments, we consider different network architectures (transformers, CNNs, and MLPs), with various activation functions.
> We also investigate different training procedures: with low-rank adapters, partial or end-to-end fine-tunings.
> We do so for many different tasks and modalities: text generation, image captioning, image-to-test generation, visual grounding, visual question answering, etc.
> Our empirical observation is that, across those setups, the **LMC is architecture-agnostic, procedure-agnostic, task-agnostic and modality-agnostic**.
>
> The main condition we require is the shared pre-trained initialization [Neyshabur2020], as emphasized in Remark 1; this "prevents the weights from diverging" (l.145) and forces them "to remain close" (l.146).
> The other condition, suggested by the literature [Li2022,Ilharco2023] and as also discussed in [R.TSwH.Q4](https://openreview.net/forum?id=lSbbC2VyCu&noteId=np0m5ACZtx), is that the architecture has enough trainable parameters. Indeed, larger networks may facilitate the orthogonality of the fine-tuned updates; then [Ilharco2023] "speculate that this [orthogonality] enables the combination of task vectors via addition with minimal interference". In conclusion, our experiments and the literature suggest **that the network design is not critical for the LMC, as long as the network is pre-trained and sufficiently parameterized**. Those constraints are arguably minimal given the predominance of the foundation model paradigm and the scaling trend in deep learning.
>
> [Neyshabur2020] What is being transferred in transfer learning? NeurIPS.\
> [Li2022] Branch-Train-Merge: Embarrassingly parallel training of expert language models.\
> [Ilharco2023] Editing models with task arithmetic. ICLR.
>
> ---
>
> ### Q3. Does the method harm the absolute quality of the produced samples? show the generated samples, and provide more evaluation
>
> **Qualitatively**, samples generated by weight interpolated models do not suffer from reduced quality. This was visible for text-to-image generation with diffusion models in Figure 12 from Appendix E.3, where we state: "we can see that all interpolated models produce images of similar quality compared to fine-tuned models". Moreover, **our anonymous website** (referenced l.856, l.1065, and l.1084), also includes generated samples for the locomotion task and for the text-to-text summarization task. For the sake of completeness, we now include **examples of generated summaries** in the Table 1 from the one-page rebuttal pdf; qualitatively, the summaries generated by interpolated models remain grammatically coherent.
>
> To **quantitatively** validate this insight, the one-page rebuttal pdf includes new plots evaluating the samples generated by RS.
>
> - Figure 1.a evaluates the generated summaries when $\lambda$-interpolating between two LLMs fine-tuned on two summary rewards. We leverage two text metrics; the first is (i) **perplexity** (exponentiated average NLL of the generated summaries) according to MLMS [Salazar2020] and GPT2 (following [Lee2021] and this [blog](https://huggingface.co/docs/transformers/perplexity)); the second is (ii) **quality**, as estimated by this [newspaper quality model](https://huggingface.co/valurank/distilbert-quality).
> - Figure 1.b evaluates the generated images when $\lambda$-interpolating between two diffusion models fine-tuned on two aesthetic rewards. We leverage two standard image metrics; the first is (i) **FID** [Heusel2018] measuring image realism; the second is (ii) **CLIPScore** [Hessel2021] measuring image-text alignment.
>
> In conclusion, we confirm quantitatively that **RS does not deteriorate quality**. More precisely, by interpolating the weights, we also interpolate the metrics; intermediate values of $\lambda$ even sometimes increase quality. We will detail this analysis in the revised paper, and would be pleased to include any other suggested quality metrics.
>
> [Salazar2020] Masked Language Model Scoring. ACL.\
> [Lee2021] Towards Few-Shot Fact-Checking via Perplexity. ACL.\
> [Heusel2018] GANs Trained by a Two Time-Scale Update Rule Converge to a Local Nash Equilibrium. NeurIPS\
> [Hessel2021] CLIPScore: A Reference-free Evaluation Metric for Image Captioning. ACL.

---

### Author Rebuttal · Authors · 2023-08-06

We sincerely thank the reviewers for their time and their insightful feedbacks. We're encouraged by the positive comments, which highlight the main features of our submission.

- (*topic*) We "address the reward misspecification problem [...] in current RLHF frameworks" (R.QPfR), a problem "that frequently arise in the emerging and important field of aligning generative models with human preferences" (R.TSwH). This "topic [is] of increasing interest and relevance to the community" (R.bXWy).
- (*methodology*) We propose rewarded soup (RS) which "involves individually training multiple networks, each assigned to a different proxy reward, and then linearly combining these networks" (R.ntSF). "The proposed idea is effective yet efficient as it does not require additional training" (R.DNE5) contrary to "to the more costly baselines" (R.bJvT) such as MORL.
- (*experiments*) Empirically, we "did a lot of experiments on different task which shows that this interpolating strategy is universal under different application scenarios, while with good performance" (R.stHc). "The paper presents interesting results for many practically relevant and useful benchmarks" (R.bXWy).
- (*theory*) "The approach is well-motivated theoretically" (R.TSwH) and "the theory part connects with experiments very well" (R.TSwH).

We have taken note of the questions and suggested weaknesses, that we directly answer in response to each reviewer.
Most of our answers are based on quotes from the main paper or the Appendix, that the reviewers might have overlooked (in particular the theoretical Appendix B.2 referenced in Section 2.2.2). In contrast, a few questions required new plots to be answered, that we gather in the one-page rebuttal pdf. Specifically:

- Table 1 shows generated samples by our interpolated models for the summarization task. This qualitative inspection is enriched by quantitative evaluations in Figure 1.a and 1.b, with general-purpose quality metrics such as perplexity for summaries and FID for image generations. We validate that the generated samples from our method do not suffer from reduced quality ([R.stHc.Q3](https://openreview.net/forum?id=lSbbC2VyCu&noteId=GlQnTpp2gl)).
- Figure 1.c and 1.d show that performances get better (Pareto-optimally) when increasing the number of averaged weights ([R.ntSF.Q2](https://openreview.net/forum?id=lSbbC2VyCu&noteId=UEM7DRMGYu)) and thus the number of training rewards ([R.QPfR.Q4](https://openreview.net/forum?id=lSbbC2VyCu&noteId=58jAIy2tXU)).
- Figure 2 quantifies the average efficiency gain from RS with regard to the MORL baseline ([R.bJvT.Q3](https://openreview.net/forum?id=lSbbC2VyCu&noteId=Hztsf2jtoo) and [R.QPfR.Q3](https://openreview.net/forum?id=lSbbC2VyCu&noteId=58jAIy2tXU)).
- Figure 3.a and 3.b plot RS's fronts over the course of fine-tuning, and validates that the LMC holds even for longer trainings ([R.TSwH.Q3](https://openreview.net/forum?id=lSbbC2VyCu&noteId=np0m5ACZtx)).
- Figure 3.c illustrates the empirical difference and the complementarity of rewarded soups and model soups ([R.DNE5.Q1](https://openreview.net/forum?id=lSbbC2VyCu&noteId=6LMJAJD6vx)).

We hope our responses clarify the expressed concerns. If there is anything else we can do to further improve our work, please let us know.

---

### Decision · Program_Chairs · 2023-09-21

**Decision:**

Accept (poster)

**Comment:**

This paper is about rewarded soup yielding Pareto optimal solutions through weight interpolation after training. As raised by many reviewers, I suggest the authors to clarify the novelty with respect to model soup and with respect to MORL in the final version.

I encourage the authors to incorporate the share concerns among the reviewers: (1) quantitative computational advantage over MORL - move figure 2 in the one page rebuttal to main paper. (2) clarify the RS targets a particular kind of reward misspecification (linear). (3) spider plots for the number of networks averaged (N>2) - move figure 1 (c,d) in the one page rebuttal to main paper.

Overall, this paper deserves acceptance at NeurIPS as a poster paper.